# MarShie: a clearing protocol for 3D analysis of single cells throughout the bone marrow at subcellular resolution

Till Fabian Mertens [1,2,12], Alina Tabea Liebheit [1,2,3,12], Johanna Ehl[1,2], Ralf Köhler[2], Asylkhan Rakhymzhan [1,4], Andrew Woehler[5,11], Lukas Katthän[6], Gernot Ebel[6], Wjatscheslaw Liublin[4], Ana Kasapi[1,7], Antigoni Triantafyllopoulou [1,7], Tim Julius Schulz [8,9], Raluca Aura Niesner [4,10,13] & Anja Erika Hauser [1,2,13] ✉

Analyzing immune cell interactions in the bone marrow is vital for understanding hematopoiesis and bone homeostasis. Three-dimensional analysis of the complete, intact bone marrow within the cortex of whole long bones remains a challenge, especially at subcellular resolution. We present a method that stabilizes the marrow and provides subcellular resolution of fluorescent signals throughout the murine femur, enabling identification and spatial characterization of hematopoietic and stromal cell subsets. By combining a pre-processing algorithm for stripe artifact removal with a machine-learning approach, we demonstrate reliable cell segmentation down to the deepest bone marrow regions. This reveals age-related changes in the marrow. It highlights the interaction between $CX_3CR1^+$ cells and the vascular system in homeostasis, in contrast to other myeloid cell types, and reveals their spatial characteristics after injury. The broad applicability of this method will contribute to a better understanding of bone marrow biology.

Long bones fulfill versatile vital functions in mammals as parts of the musculoskeletal system and in their role of housing the bone marrow (BM), the site of hematopoiesis. BM is a very heterogeneous tissue, comprising a variety of different hematopoietic cell types, as well as mesenchymal stromal cells, and a complex vascular system consisting of different vessel types[1–3]. Even in steady-state homeostasis, the marrow vasculature shows a high degree of morphological alterations over time[4]. Mesenchymal stromal cells and vessels form a three-dimensional scaffold, which hosts and actively supports hematopoietic cells, thereby contributing to hematopoietic stem cell maintenance and the differentiation into various blood cell lineages[5]. The marrow also becomes increasingly recognized as a site where immunological

[1]Department of Rheumatology and Clinical Immunology, Charité - Universitätsmedizin Berlin, corporate member of Freie Universität Berlin and Humboldt-Universität zu Berlin, 10117 Berlin, Germany. [2]Immune Dynamics, Deutsches Rheuma-Forschungszentrum (DRFZ), a Leibniz Institute, Charitéplatz 1, 10117 Berlin, Germany. [3]Institute of Chemistry and Biochemistry, Department of Biology, Chemistry and Pharmacy, Freie Universität Berlin, Berlin, Germany. [4]Biophysical Analytics, Deutsches Rheuma-Forschungszentrum (DRFZ), a Leibniz Institute, Charitéplatz 1, 10117 Berlin, Germany. [5]Berlin Institute for Medical Systems Biology, Max Delbrück Center for Molecular Medicine, 10115 Berlin, Germany. [6]Miltenyi Biotec B.V. and Co. Bertha-von-Suttner-Straße 5, 37085 Göttingen, Germany. [7]Innate Immunity in Rheumatic Diseases, Deutsches Rheuma-Forschungszentrum (DRFZ), a Leibniz Institute, Charitéplatz 1, 10117 Berlin, Germany. [8]Department of Adipocyte Development and Nutrition, German Institute of Human Nutrition (DIfE) Potsdam-Rehbruecke, 14558 Nuthetal, Germany. [9]German Center for Diabetes Research (DZD), 85764 Munich-Neuherberg, Germany. [10]Dynamic and Functional in vivo Imaging, Veterinary Medicine, Freie Universität Berlin, Berlin, Germany. [11]Present address: Janelia Research Campus, Howard Hughes Medical Institute, Ashburn, VA 20147, USA. [12]These authors contributed equally: Till Fabian Mertens, Alina Tabea Liebheit. [13]These authors jointly supervised this work: Raluca Aura Niesner, Anja Erika Hauser. ✉e-mail: anja.hauser-hankeln@charite.de

memory is maintained[6,7] further confirming its important role in systemic immune protection. On the other hand, inflammatory changes of BM stromal cells promote hematologic tumor outgrowth and counteract anti-tumor therapy[8]. Recent reports have indicated that vessels[9] and stromal cells in BM undergo age-dependent changes[10] and are probably more heterogeneous than previously thought. However, many questions regarding the extent and phenotype of individual stromal, vascular and immune cells in BM, under homeostasis but also during perturbations such as injuries, remain unanswered. This is partly due to a lack of suitable methods to quantitatively analyze spatial relationships between those cell compartments in 3D.

Light sheet fluorescence microscopy (LSFM) has the potential to answer some of those questions, as it allows the analysis of cells and structures within whole, cleared organs. For example, the application of LSFM after ethyl cinnamate (ECI) clearing in bones has helped to identify vascular structures termed transcortical vessels[11] and, very recently, the description of lymphatic vessels within BM[3] led to a change in perception of the vascular compartment in bones. Adding to that, other clearing protocols have been applied to visualize the BM. Bone CLARITY elucidated the spatial distribution of osteoprogenitors in proximity to the cortex[12]. PEGASOS[13] and BoneClear[14] visualized the network of peripheral neurons in long bones and PACT[15,16] was used to visualize immune cells in the BM of rodents. Gorelashvili et al. used benzyl alcohol/benzyl benzoate (BABB) clearing to determine the size and quantity of megakaryocytes in parts of the BM[17]. The same group recently developed an image segmentation pipeline to analyze megakaryocyte localization in BM image stacks[18], the results of this approach were subsequently used to model the distribution of those cells in the whole marrow[19].

Evidently, these protocols work well for analyzing smaller volumes, usually close to the cortex, and for visualizing structures significantly larger than most individual hematopoietic cells. However, a method to fully preserve and depict the three-dimensional tissue architecture of whole murine long bones, in particular large ones such as the femur, including its deep marrow, at single- and subcellular resolution, has not been established yet. Optical clearing of BM itself presents unique challenges, mainly related to the specific characteristics of this tissue. Most importantly, the fragile BM structure within the hard cortex needs to be preserved. The BM is traversed by many large vessels that may collapse during some clearing protocols, causing a reduction in volume and change in spatial relationships of single cells. Other protocols expand the marrow volume, which is particularly a problem for injury model analysis as the marrow leaks out of the marrow cavity, this also causes distortions of the tissue texture. In addition to tissue stabilization, the BM within whole, intact bones is difficult to image due to several reasons: the presence of pigments, especially hemoglobin, poses a challenge, whilst high refractive variability in cortex, but also in the marrow lead to strong light scattering and absorption of radiation. Here, we overcome those limitations by combining a clearing method optimized for imaging at cellular and subcellular resolution throughout the entire marrow with a post-processing approach, which renders the data suitable for AI-based image analysis. Our method reveals age-dependent changes in the vascular compartment of bones and provides quantitative spatial information of immune cell subsets at a single cell level, throughout the BM, under physiological and pathological conditions.

## Results

### MarShie - A protocol for bone marrow clearing, optimized for both tissue preservation and transparency

We aimed to effectively clear whole murine femurs in tissue homeostasis and during regeneration after injury, down to the deepest BM regions, to enable imaging at single cell and even subcellular resolution. Therefore, we tested available tissue clearing protocols specifically designed for bone clearing, i.e., PEGASOS[13], ECI[20], FDISCO[21] and Bone CLARITY[12]. Albeit their sophisticated clearing abilities and preservation of the different fluorescence signals in deep BM regions, we found them unable to reliably preserve the fragile soft BM tissue architecture. Our attempts stress that it is highly challenging to retain the tissue architecture of the delicate, interior marrow, especially in large volumes such as the femoral bone.

We present here a protocol specifically addressing those challenges (Fig. 1A), that allows the generation of images adequate for semiautomatic quantification of cellular and subcellular signals. To provide a stable, yet flexible scaffold to the marrow tissue, we base our protocol on the introduction of a durable SHIELD polyepoxide fixation[22] and adapt it for the BM. We remove carbonated hydroxyapatite, the main inorganic component of bone, with a mild decalcification step by chelation with ethylene-diamine-tetra-acetic acid (EDTA). By using an electrokinetic approach[23], we enhance the homogenous de-lipidation of the marrow. Our protocol improves decolorization efficiency, by combining N,N,N′,N′-Tetrakis(2-Hydroxypropyl)ethylenediamine (Quadrol)[24] based decolorization with the zwitterionic detergent 3-[(3-cholamidopropyl)dimethylammonio]−1-propanesulfonate (CHAPS)[25]. This protocol results in complete clearing of voluminous long bones (Fig. 1B), exemplified here by the mouse femur. We demonstrate that our clearing approach is able to preserve the 3D fragile marrow tissue architecture in whole long bones without any notable tissue swelling or shrinkage (Fig. 1C, D), a prerequisite for subsequent quantitative spatial analyses of single cells. Our protocol prevents the detachment of the marrow from the cortex at endosteal sites and no other tissue artifacts such as ruptures are introduced in the marrow. To reflect the preserved structure of the BM, we termed the protocol MarShie, an acronym for marrow shield. To quantitatively assess tissue transparency, we performed spectroscopic analyses and achieved >90% maximum transmission at 785 nm. (Fig. 1E). As expected, the transmission of radiation in the visible spectrum through the samples improves with incrementing wavelength (Fig. 1F).

In short, MarShie achieves a high degree of transparency while preserving the femoral BM architecture in all regions of the femoral long bone, which is the prerequisite for imaging with subcellular resolution in deep BM regions.

### MarShie preserves various fluorescence signals throughout the murine femoral bone marrow and allows for imaging at high signal-to-noise ratio with subcellular resolution

We confirmed the compatibility of MarShie with the preservation of various fluorescent signal types. We used CX$_3$CR1-GFP x Cdh5-tdTomato/histone-GFP reporter mice that express GFP in CX$_3$CR1$^+$ cells, spatially separated from histone-linked GFP in nuclei of the endothelial cells, and tdTomato (tdTom) in endothelial cell membranes via Cdh5-linkage[26,27]. Fluorescent labeling for LSFM included (i) endogenous GFP, expressed in CX$_3$CR1$^+$ myeloid cells and histone-linked GFP in endothelial nuclei (ii) the red fluorescent protein tdTom driven by the Cadherin 5 (Cdh5) promoter (iii) a whole-mount stained signal using nanobodies against GFP, conjugated to Atto-647N, and (iv) intravenously injected antibodies against CD31, coupled to Alexa647 (Al647). To boost the signal of GFP and its variants to longer wavelengths, we implemented a nanobooster-staining step (Fig. 1A) that we adapted from the mild vDISCO protocol[28]. Using our pipeline, we detect high fluorescence intensities of GFP and tdTom, CD31-Alexa647 and anti-GFP-Atto-647N down to the deepest BM regions, which are marked by the presence of the central sinus (Fig. 2A). In addition, MarShie preserves anatomical landmarks such as the endosteal lining, transcortical vasculature and the deep BM sinus volume.

We determined the signal-to-noise ratio (SNR) for different fluorescence signals in endosteal and deep BM femoral regions in the diaphysis (Supplementary Fig. 1A–D). To that end, for the GFP signal we measured an SNR of 7.4 at the endosteum, and an SNR of 4.0 in deep BM regions. The Cdh5-tdtTom$^+$ endothelial cells retained a high

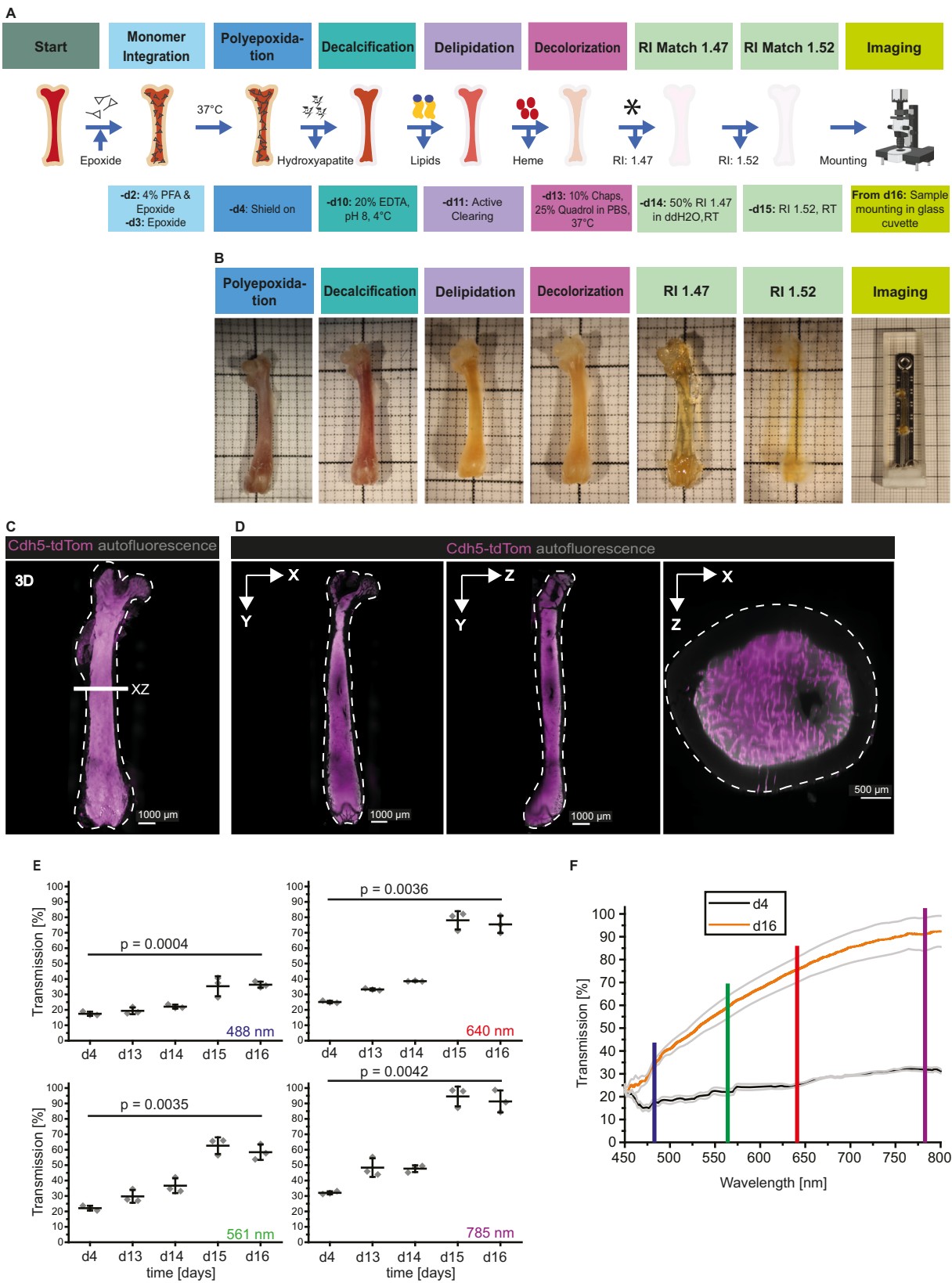

SNR, with average values of 12.7 and 8.5 in endosteal as well as deep BM regions, respectively. Following nanobody staining, we detected the highest SNR in the far-red emission wavelengths of Atto-647N, amounting to 22.9 at endosteal and 20.3 in the deep marrow region (Fig. 2B). The preservation of signal intensity becomes evident in representative z-slices from the deep BM of the tdTom and anti-GFP-Atto-647N signal with three prominent straight vessels, resembling arteries (Fig, 2C, blue inset) and GFP-expression in both endothelial nuclei as well as $CX_3CR1$-GFP$^+$ myeloid cells (Fig. 2D, orange inset). Line plots of the Cdh5-tdTom (Fig. 2E) and anti-GFP-Atto-647N (Fig. 2F)

**Fig. 1 | MarShie enables full preservation of the marrow structure and effectively renders the tissue transparent. A** Schematic representation of the MarShie protocol. Asterisk (after day 13) indicates time point for optional immunolabeling/ nanobooster. RI: refractive index. Illustration created with BioRender.com. **B** Representative images of the murine femur after each MarShie working step. Images are taken outside of the respective medium, except for day 15 and 16, when bones are imaged in the final matching solution. **C** 3D-Reconstruction of a murine femur after MarShie clearing (Cdh5-tdTom reporter mouse, 20 weeks). Cdh5-tdTom in magenta, autofluorescence in gray. White dashed lines in (**C**) and (**D**) indicate the bone outlines. **D** Orthogonal slice views (20 μm thick) in xy, yz and xz. White line in (**C**) depicts the position of the xz-orthogonal plane at the diaphysis.

**E** Optical transmission of the diaphysis at 488 nm, 561 nm, 640 nm and 785 nm (±20 nm) after respective clearing steps of MarShie. Rhomboid data points depict mean of three individual measurements. Data (mean ± SD) of three different femurs (C57Bl/6J, 16-weeks and 11-weeks old) are analyzed by unpaired, two-tailed *t* test between day 4 and day 17. Source data are provided as a Source Data file. **F** Comparison of optical transmission over a range of excitation wavelengths (mean ± SD of three femurs from two individual mice) of the diaphysis between day 4 (black) and day 17 (orange). Vertical lines represent excitation laser lines of the light sheet microscope, 488 nm in blue, 561 nm in green, 640 nm in red and 785 nm in purple. Source data are provided as a Source Data file.

signals highlight the clear separation of vessel and immune cell signals from the surrounding background (Supplementary Fig. 1E). Next, we tested the ability of our MarShie protocol to preserve endogenous fluorescence for an extended time after clearing. For Cdh5+ tdTom fluorescence, we demonstrate a nearly constant average signal-to-background ratio (SBR) for at least six months (Fig. 2G). In Fig. 2H, the SBR of the Cdh5-tdTomato signal is depicted by color-coding at various time points after clearing, in representative deep marrow images.

The protocol is compatible with a wide palette of other fluorescent proteins, labeling hematopoietic cells as well as stromal subsets. Examples include the mTmG reporter mouse strain for cell-type specific labeling, as it switches from ubiquitous, membrane-bound tdTomato expression (mT+) to membrane -bound GFP expression (mG+) following Cre recombination while the surrounding, non-recombined cell types retain red fluorescence. The mTmG mice were crossed to the adipocyte-specific Adiponectin-Cre strain[29]. Here, mG+ adipocytes were readily detected within the BM, whereas the surrounding hematopoietic tissue remained mT+ (Supplementary Fig. 2A).

For labeling B lymphocytes, we used tandem red fluorescent protein (tdRFP[30]) which we imaged in the BM of CD19-tdRFP fate map reporters[4]. This way, we could clearly visualize the distribution of B lymphocytes in the BM, as shown in Supplementary Fig. 2B and Supplementary Movie 1. B lymphocytes are produced in the BM, but mature B cells circulate in the blood. To find out whether the B cells locate in the BM parenchyma or within vessels, we explored the possibility to perform in vivo stainings for vasculature, as well as for hematopoietic cells. Vessels were labeled by injecting an antibody directed against CD31, labeled with Alexa Fluor 647 into CD19-tdRFP mice (Supplementary Fig. 2B). The data show that the B lymphocytes are predominantly located in the BM parenchyma, thereby appearing to be evenly distributed.

As demonstrated in Supplementary Fig. 2C, D, the protocol also preserves the fluorescence of monomeric fluorescent Kusabira Orange (mKO2) and Azami Green (mAG) in the BM. This allows labeling nuclei in the G1 and S/G2/M phases of the cell cycle, respectively[31]. Both mKO2+ and mAG+ cells are individually distributed in the BM, with no obvious preference for specific areas.

Next, we tested whether hematopoietic cells in the BM could also be stained with fluorescently labeled antibodies. For labeling myeloid cells, we injected an anti-CD169 antibody conjugated to eFluor (eF) 660 into Cdh5-tdTom/Histone-GFP mice, This approach results in labeling of cells with a reticular morphology, which localize in parenchymal areas of the BM. (Supplementary Fig. 2E and Supplementary Movie 2). The antibody-mediated staining can be observed with consistent intensity down to the deep BM.

Label-free detection of collagen using second harmonic generation (SHG) is an important advantage when 2-photon imaging of bones is performed. It could also be useful for imaging cleared bones. This prompted us to test whether our protocol preserves collagen structures. Indeed, we can detect a strong SHG signal in the cortex of MarShie-cleared bones, in addition to vessels in the cortex and marrow of Cdh5-tdtTom mice (Supplementary Fig. 3 and Supplementary Movie 3). The maximum imaging depth in cleared bones of the TdTom

signal amounts to 400 μm using state-of-the-art 2-photon microscopy, confirming LSFM as the optimal solution for femoral BM imaging in toto.

Taken together, our MarShie protocol allows the detection of various fluorescence signals and different cell types within the deep BM, clearly separating the cellular signals from the surrounding background with a high SNR. The preservation of collagen structures enables label-free visualization using SHG, and the possibility to perform antibody-based staining of hematopoietic as well as stromal cell types further reduces the dependence on fluorescent reporter mouse strains, extending the applicability of the protocol.

## Tissue clearing and image pre-processing permit visualization of the complete vascular system and stromal cell morphology throughout the bone marrow

A detailed analysis of the murine BM vasculature in toto has remained challenging, in part due to the high levels of autofluorescence and light absorption characteristic for this tissue. Using MarShie, we are able to image the vascular system of the entire murine femoral bone (Fig. 3A and Supplementary Movie 4). Slicing the bone volume from anterior to posterior highlights that there is no detectable decrease of the vascular Cdh5-tdTom signal in deep z-stacks (Fig. 3B). Due to MarShie's preservation of soft tissue we are now able to demonstrate the dense vascular network of the whole murine femur in morphological detail (Fig. 3C and Supplementary Movie 5), including the central sinus (Fig. 3D, 1), the sinusoidal network (Fig. 3D, 2), straight vessels that resemble arteries[32] (Fig. 3D, 3) and transcortical vessels (Fig. 3D, 4). Furthermore, we are able to detect the histone-linked GFP expression of single endothelial cell nuclei, after nanobody-dependent signal boosting (Fig. 3E). Single endothelial nuclei are clearly separated from each other, demonstrating the ability of MarShie to achieve subcellular resolution, even inside the deep marrow.

Within a representative diaphyseal volume, we demonstrate that we can semi-automatically detect individual endothelial nuclei using machine-learning algorithms. To perform machine-learning based segmentation, we pre-processed the images of the Cdh5-tdTom signal to suppress striping artifacts present in the marrow regions of the raw data. We discovered that the directions associated with the striping artifacts correspond to the illumination pattern geometry (Supplementary Fig. 4A), convolved with the beam divergence and light sheet width distribution, which define the spatial resolution of our microscope. To eliminate only the artifact frequencies while preserving sample details, we created a directional filter in Fourier space (fast Fourier transformation, FFT), taking advantage of the known direction of illumination (Supplementary Fig. 4A). This filter replaces frequencies along the stripe directions with a value of one and applies Gaussian blurring to adjacent areas in the FFT image. The algorithm succeeds to suppress most stripes in the images (Supplementary Fig. 4B–E), with the only exception of hardly visible, low amplitude higher harmonics. Compared to other published methods such as combined wavelet-Fourier filtering[33] or non-subsampled contourlet transform[34], our algorithm yields comparable or even better results in the reduction of artifacts for our application, whilst preserving

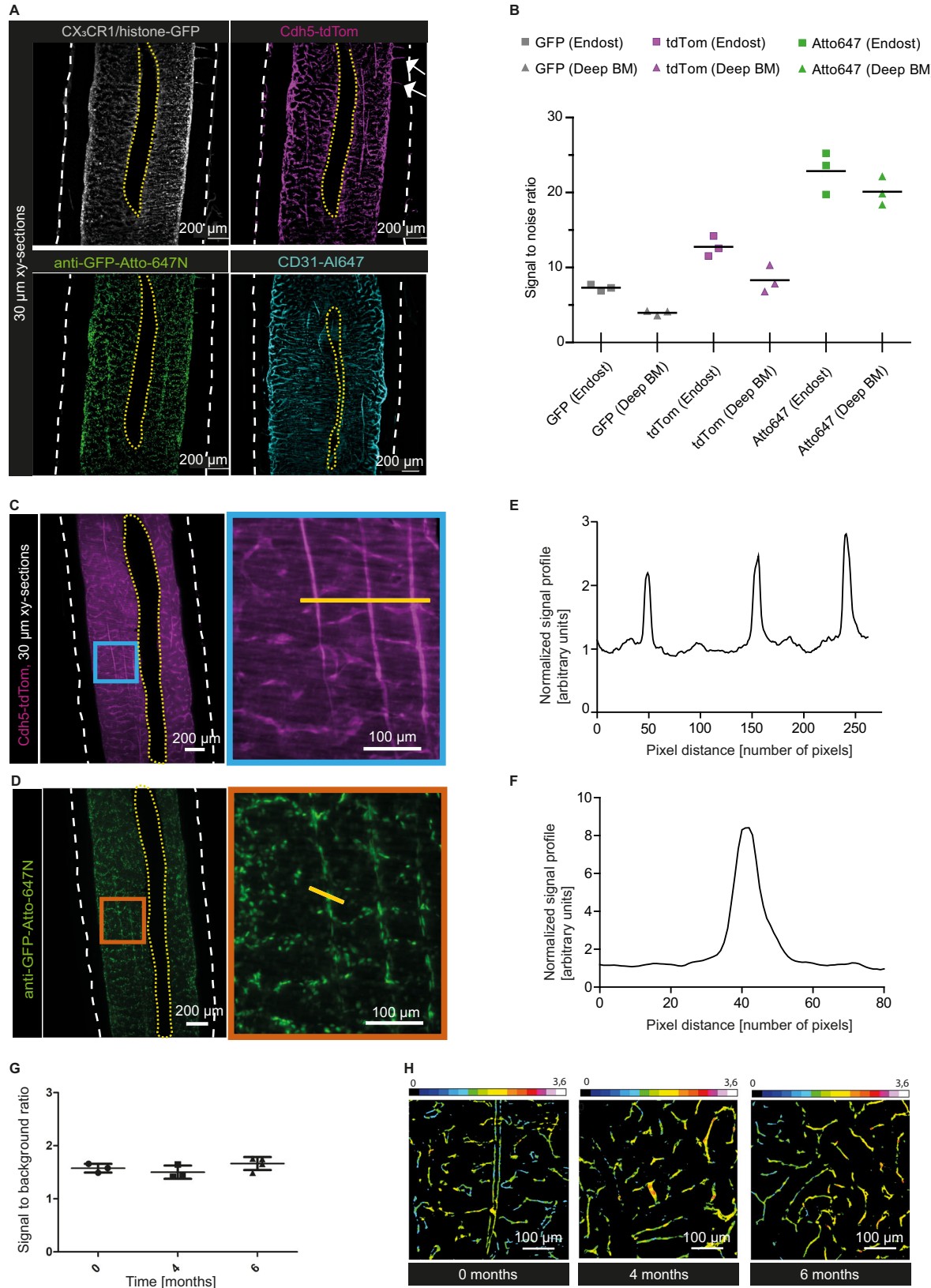

contrast and spatial resolution, and requiring less processing time. By Gaussian fitting the first derivative of the line profiles crossing the same structure in the BM (Supplementary Fig. 4F) we demonstrated no loss of spatial resolution for any de-striping approach we applied (raw image: $2.1 \pm 0.2\,\mu m$; our directional FFT approach: $2.2 \pm 0.2\,\mu m$; combined wavelet: $2.1 \pm 0.1\,\mu m$; non-subsampled contourlet: $2.3 \pm 0.2\,\mu m$, Supplementary Fig. 4G). The pre-processing results in images that can be subjected to semi-automated segmentation of the vascular structures (Fig. 3F and Supplementary Movie 6). Employing user-based annotations, followed by pixel-wise semantic segmentation

**Fig. 2 | Quantification of fluorescent signals at endosteal and deep BM regions.** **A** Representative xy-orthogonal slices (30 μm thick) of deep BM regions. White arrows point at exemplary transcortical vessels. CX$_3$CR1/histone-GFP signal in gray (12 μm background subtracted), Cdh5-tdTom signal in magenta (20 μm background subtracted), anti-GFP-Atto-647N signal in green (12 μm background subtracted), CD31-Al647 signal in cyan (25 μm background subtracted). White dashed lines indicate bone outlines, yellow dashed lines indicate the central sinus. **B** Signal-to-noise ratio (SNR) of endogenous GFP signal, endogenous tdTom signal and anti-GFP-Atto-647N signal in endosteal and deep BM regions. Data show mean (*n* = 3) of individual, young mice and data points are the mean of three consecutive z-slices of one femur each. Source data are provided as a Source Data file. **C** Representative z-slice (30 μm thick) showing the Cdh5-tdTom signal in magenta within the deep BM. Right: Zoom-in at the blue framed region shows three straight-running vessels. White dashed lines indicate bone outlines, yellow dashed lines indicate the central

sinus. Yellow line indicates region of the signal profile in (**D**). **D** Line plot shows the tdTom signal profile normalized to the background. Source data are provided as a Source Data file. **E** Representative z-slice (30 μm thick) within the deep BM, which shows the CX$_3$CR1/histone-GFP signal nanoboosted by anti-GFP-Atto647N in green. Right: Zoom-in at the orange framed region. White dashed lines indicate the bone outlines, yellow dashed lines indicate the central sinus. Yellow line marks the pixels used to determine the signal profile in (**F**). **F** Line plot shows the anti-GFP-Atto647N signal profile normalized to the background. Source data are provided as a Source Data file. **G** Signal-to-background ratio (SBR) of the tdTom signal 0, 4 and 6 months after clearing with MarShie. Data are mean ± SD of *n* = 3 femur (0 and 4 months) and *n* = 4 femur (6 months). Source data are provided as a Source Data file. **H** Color-coded SBR (intensity indicated by the rainbow scale) from Cdh5-tdTom· up to 6 months after clearing.

from LABKIT, a random forest classifier optimized for fast implementation and memory efficiency[35], a quantification of vascular volume within the whole BM was performed. We first validated the performance of the classifier by comparing it to annotations performed by trained raters in the different BM regions (Supplementary Fig. 5A) using object intersection over union (IoU[36]) as an overlap-based metric for segmentation assessment. This yielded mean values of 0.65 ± 0.04 (s.d., range from 0.6 to 0.69) and 0.68 ± 0.06 (s.d., range 0.59 to 0.74) for endosteal and deep marrow, respectively (Supplementary Fig. 5B). Interrater IoU values for endosteal marrow segmentation ranged between 0.6 and 0.72 (Supplementary Fig. 5C). Thus, the performance of the segmentation algorithm is similar to that of the individual raters.

As evident from overlay images, our de-striping tool[37] is necessary to remove stripe artifacts that are otherwise misclassified as vessels (Supplementary Fig. 6A–C). Thus, it enables a more accurate analysis of tissue structures. The subcellular resolution in comparison with our segmentation allows us to register individual endothelial nuclei of both arterial and sinusoidal vessels as well as of the central sinus (Fig. 3G). In total, we detect 16.904 endothelial nuclei (16.465 within the cylindrical diaphyseal BM volume of the sinusoidal and arterial vessels plus 439 nuclei of the central sinus). This corresponds to 39.960 nuclei/mm³ BM, assuming a uniform distribution in the defined BM volume.

In addition to the vascular compartment, we use MarShie to study the mesenchymal network of the femoral BM (Fig. 3H). In Prx1-YFP fate map reporter mice, we detect YFP⁺ osteocytes as well as YFP⁺ mesenchymal stromal cells (MSCs), which closely envelope transcortical vessels (Fig. 3I, 1). Within the BM, we can discriminate larger (about 25 μm) and smaller, round YFP⁺ MSCs, reflecting heterogeneity among the MSC population, next to their fine extensions, which connect individual MSCs with each other (Fig. 3I, 2, 3). These long projections extend in all directions, thus the dimensions of single MSCs cannot be fully appreciated by 2-dimensional microscopy. As demonstrated in close-up images of Prx-fate map reporter mice injected with anti-CD31, we can capture the extensions of MSCs in all directions (Fig. 3J), revealing a network of stromal cells which forms in addition to the vascular network (Fig. 3K, left image). Notably, the high quality of the images facilitates segmentation of stromal cells, allowing quantification of the stromal network in 3D (Fig. 3K, right image).

## MarShie reveals relevant age-related changes in the bone marrow vasculature

Age-related changes of the BM vasculature might result in functional consequences for tissue homeostasis and regenerative capacity[2]. Comparing the femoral marrow vasculature between young (20–25 weeks) and aged (84–86 weeks) mice, our 3D data highlight a significant volumetric deterioration of the central venous sinus in the latter (Fig. 4A and Supplementary Movies 7 and 8). For robust

quantification, we performed a manual surface segmentation of the central sinus volume in both age cohorts (Fig. 4B). In the aged femoral mid-diaphyseal volume, the central sinus volume amounts to 0.075% (±0.016% SEM) of the total BM volume, whereas in young individuals, the central sinus occupies around 4.28% (±0.95% SEM) (Fig. 4C).

Described in mice and humans, age-induced cortical porosity in female bone structure is a common phenomenon[38,39]. In young mice, transcortical vessels traverse the cortex and connect the BM to the periosteal circulation, as shown in the murine tibia[11]. Our data confirm these findings in the femur, as only very few transcortical vessels are present in the femoral whole-mounts of aged, female mice. However, we discovered volumes inside the cortical bone of aged mice that resemble the BM in the medullary cavity (Fig. 4D). Inside these ectopic marrow islands, a highly convoluted vessel network is abundant, similar to the sinusoidal network in the marrow (Fig. 4E, F). This network communicates with the BM circulation, as well as with the periosteal vessel network, via short, stalk-like connecting vessels (Fig. 4E, blue arrowheads and Supplementary Movie 9). To assess the extent of these areas, we compared the non-mineralized intracortical volumes within the distal diaphysis between aged and young mice. Random forest-based pixel classification was used to separate the non-mineralized intracortical volume from mineralized areas within the cortex, followed by automated surface rendering to reconstruct the non-mineralized intracortical volume (Fig. 4G). We find a significantly greater intracortical non-mineralized volume (comprising intracortical vessels as well as marrow regions) in the bones of the aged mice, namely 3.42% (±0.85% SEM) versus 1.01% (±0.36% SEM) of the total cortex volume (Fig. 4H). While in young mice, these volumes almost exclusively constitute the transcortical vessel network, they appear more like ectopic marrow islands residing inside the thinned cortex of aged mice.

## Distinct 3D spatial distribution of different myeloid cell types in the femoral marrow

We used CX$_3$CR1-GFP x Cdh5-tdTom/histone-GFP reporter to analyze both the tdTom⁺ vessels and the GFP⁺ myeloid cells. The GFP⁺ signal was enhanced with anti-GFP-Atto647N nanobooster. Adding this approach to our clearing pipeline, we can visualize all CX$_3$CR1-GFP⁺ myeloid cells throughout the whole-mount down to deep BM regions (Fig. 5A). Besides the single cell bodies, we can visualize their detailed morphology including their fine dendrite-like extensions (Fig. 5B). In both endosteal and deep marrow regions, we are able to automatically detect straight, artery resembling vessels (Fig. 5C, 1) as well as the sinusoidal network (Fig. 5C, 2) converging in the central sinus (Fig. 5D). By visual inspection, CX3CR1-GFP⁺ cells appear in close contact with the vasculature (Fig. 5E, F). When applying our machine learning-based pipeline, we can reliably segment CX$_3$CR1-GFP⁺ myeloid cells, also in deep marrow regions (Fig. 5F and Supplementary Movie 10). Similarly, this approach achieves reliable segmentation of myeloid cells stained with anti-CD169-eF660 (Fig. 5G, H). The quantification of myeloid cells

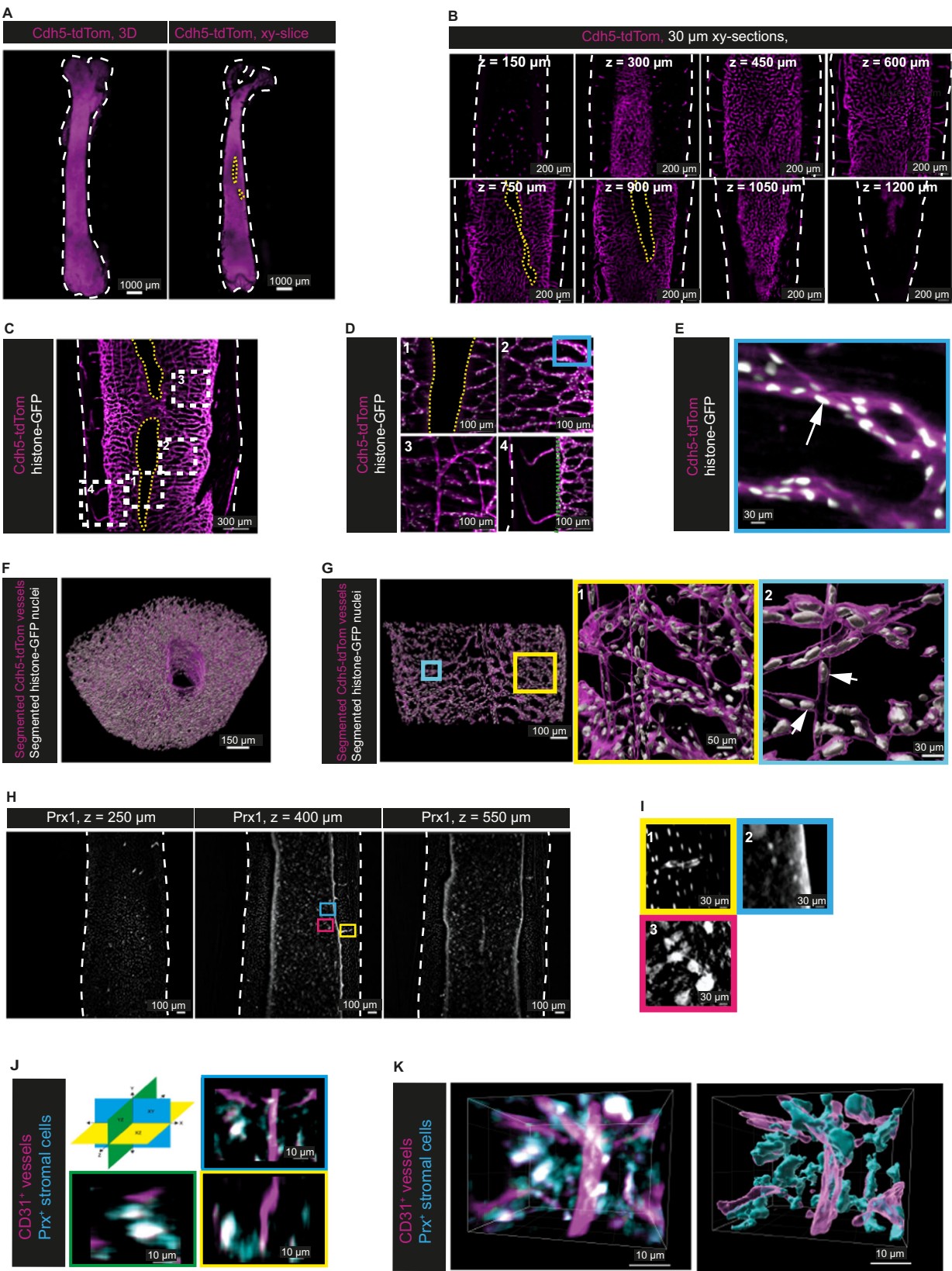

throughout the cylindrical marrow volume results in 11042 $CX_3CR1$-$GFP^+$ cells (3008 cells/mm³ BM) and 9508 $CD169$-$eF660^+$ cells (2552 cells/mm³ BM). The segmentation confirms that $CX_3CR1$-$GFP^+$ cells do not distribute randomly throughout the BM, but rather localize in close proximity to vessels. In contrast, $CD169^+$ myeloid cells appear distributed over the entire parenchyma.

**MarShie reveals three-dimensional insights into regeneration occurring in bone injury models**

In bone injury models, we previously defined a role for myeloid cells in promoting vessel sprouting into the fracture gap[40] however, the imaging technologies used did not allow a 3D analysis of the whole fracture gap. Here we take advantage of the fact that MarShie preserves the

**Fig. 3 | Imaging and pre-processing of the BM stromal compartment throughout the whole femur and at subcellular resolution. A** Left: 3D-reconstruction of a murine femur after clearing (Cdh5-tdTom reporter mouse, 20 weeks). Right: xy-orthogonal slice (30 μm) within deep bone marrow. Cdh5-tdTom signal in magenta. White dashed lines in (**A**–**D**) indicate bone outlines, yellow dashed lines indicate the central sinus. **B** Representative xy-orthogonal slices (30 μm) throughout the bone marrow (every 150 μm in z). Cdh5tdTom signal in magenta, $n = 4$ analyzed in independent experiments. **C** Exemplary xy-orthogonal slice (50 μm thick) within diaphyseal deep marrow (Cdh5-tdTom/histone-GFP mouse, 20 weeks), rectangles indicate zoom-ins. Cdh5-tdTom signal in magenta, histone-GFP[+] in white, $n = 4$ analyzed in independent experiments. **D** Zoom-ins of (**C**) show morphologically divergent vessels: (1) central sinus-lining endothelium, (2) sinusoidal network, (3) straight arteries, (4) transcortical vessels (endost marked by green line). **E** Zoom-in of blue rectangle in d2 marks histone-GFP signal of an endothelial nucleus (white arrow). **F** 3D-segmentation of Cdh5-tdTom[+] vasculature

and histone-GFP[+] endothelial nuclei within a diaphyseal volume (500 μm in y, without cortex). **G** xy-orthogonal enlarged slice of the 3D-segmentation in (**F**), rectangles indicate zoom-ins. (1) Zoom-in highlights 3D-segmentation of arterial-like straight vessels and sinusoids, (2) Zoom-in highlights segmented endothelial nuclei (white arrows). **H** xy-orthogonal slices (12 μm thick) of two independent experiments (Prx1-YFP fate map reporter mouse, 23 weeks old). White dashed lines in (**A**–**D**) indicate bone outlines. Blue, magenta, and yellow frames mark zoom-ins. **I** Zoom-ins of the Prx1-YFP signal from different MSC types: (1) Osteocytes in the cortex and MSCs enveloping a transcortical vessel, (2) Endosteal signal and small, round MSCs, (3) large MSCs with extensions. **J** 3D representation of stromal cells (cyan) in Prx1-RFP fate map reporter mice in combination with vessel labeling by injection of fluorescently labeled anti-CD31 antibodies (magenta) highlights the dimensions of single stromal cells in x,y and z as indicated in the schematic. **K** Stromal and vascular networks in Prx1-RFP mice injected with anti-CD31 antibodies (left) can be segmented (right).

tissue even in bone injury models. We analyze bones at day 1, 2, 3 and 7 after a drill hole injury, and a full osteotomy model at day 14 after surgery (Fig. 6A). Those different procedures resemble intramembranous and endochondral bone regeneration, respectively (Fig. 6B). In our analyses of the drill hole defects, we observe an early migration of CX3CR1[+] myeloid cells and subsequent clustering of these innate immune cells around the drill hole borders, starting at day two post-surgery (Fig. 6C and Supplementary Movie 11). At this time point, the cells are already present at the inner interface of the injury, essentially sequestering the damaged volume from the surrounding tissue, and preceding the vascularization process, which in our experiments is observed from day 3 onwards (Fig. 6C). The CX3CR1[+] myeloid cells consistently display ramified extensions and appear in close contact with each other, effectively touching neighboring cells. Whereas during the early time points after injury, they lose their close contact to the vasculature and appear dispersed in the marrow, they realign with endothelial cells by day 7 after injury. In the 3D segmented data (Fig. 6D), we are able to identify an increase in myeloid cell numbers per mm[3] of drill hole volume from 737 at day 1 to 25.100 at day 7 post injury. Likewise, we are able to quantify the process of vascularization, with the vessel volume increasing from 0.04% of the total drill hole volume at day 1 to around 11.4% at day 7 (Fig. 6E). In the osteotomy model, CD31[+] vessels, stained via intravenous antibody staining, sprout into the osteotomy gap after fracture. In all the individuals ($n = 5$) studied here at day 14, those vessels consistently are connected to the vasculature located at the medial periosteum, which suggests spatially regulated vascular sprouting (Fig. 6F). When compared to vessels in other BM regions, they express higher levels of CD31, identifying them as type H vessels[40].

## Discussion

MarShie is a tissue-clearing pipeline specifically suited for deep bone and BM imaging in intact murine samples. Using this pipeline, we are able to render large whole-mount murine long bones fully optically transparent. Concurrently, we maintain the 3D tissue architecture and retain the morphology of tissue compartments, particularly the marrow, which is susceptible to distortion due to its fragile structure. Hence, the stabilization of the sinusoidal cavities allows us to detect and quantify an age-related shrinkage of the central sinus. MarShie also serves to fix both hard and soft tissue components in bone injury models, i.e., drill holes and osteotomies. For this application, other clearing protocols result in substantial deformation and even leakage of the marrow out of the damaged bone cortex, particularly during very early time points after injury, when profound tissue destruction and subsequent hematoma formation occur.

In addition to tissue preservation, BM, when analyzed in whole bone, presents other challenges for LSFM. It is known that scattering and absorption of radiation cause exponential attenuation of both excitation[41] and emitted light, e.g., fluorescence, in tissues. The high

hemoglobin concentration in the hematopoietic, highly vascularized BM plays a major role in this context, as this blood pigment strongly absorbs light and limits the maximum optical penetration depth in tissue[42]. By introducing an enhanced decolorization step after active de-lipidation, we minimize light absorption and, by that, attenuation of radiation throughout the BM. All reagents used in the MarShie protocol are non-toxic. To accelerate the sample processing, we rely on an active clearing device. However, it is also possible to perform passive de-lipidation, which makes the method available to a wide range of researchers.

During imaging, light scattering which occurs within the cortex and marrow, as well as at the interface of those compartments, i.e., the endosteum, leads to a pattern of striped artifacts in the images. This effect becomes further enhanced by excitation light absorption. Various customized optical beam-path designs have been proposed and applied by expert labs to reduce the generation of striped artifacts in LSFM[43]. However, such sophisticated, customized solutions are not readily accessible to a broad research community. The orientation of the quasi-periodic striped artifacts is imposed by the alignment of the six excitation beam-paths of the microscope setup. We expect higher scattering in the cortex than in the marrow because larger abrupt RI mismatches on a small nanometer scale are present in those regions. We observed that these stripe artifacts do not affect the mean signal values and the median signal distribution in structures larger than the main period of the striped pattern, i.e., the distance between two neighboring maxima. However, in order to optimize the image quality for performing high-precision image segmentation of the Cdh5-tdTom[+] vasculature, we apply a custom-tailored directional FFT de-striping algorithm, taking the orientation of the beam-paths into account. This approach efficiently reduces stripe artifacts, preserves contrast and spatial resolution and requires less processing time, compared to existing de-striping algorithms using wavelet or contourlet transforms[33,34]. Thus, a reliable separation of fluorescence signal and background is achieved.

Machine-learning based segmentation analysis to identify single cells within the marrow of femoral bone opens up roads to characterize and quantify cell types and their spatial organization in BM niches. Segmentation algorithms for the quantitative analysis of vascular trees in LSFM data have been described for various organs, but not for the BM[44]. We demonstrate that removal of artifacts, such as de-striping, is crucial to achieve optimal segmentation results. In addition, the segmentation algorithm needs to be carefully chosen and validated. As previously proposed, we have used IoU as a measure for the segmentation performance of our algorithm compared to trained raters. This approach yielded similar IoU values to published algorithms[36,45]. However, the performance of automated bioimage analysis highly varies due to heterogeneity of the input data, and is therefore difficult to compare in distinct data sets[46]. Consequently, we decided to assess the performance of our approach in comparison to

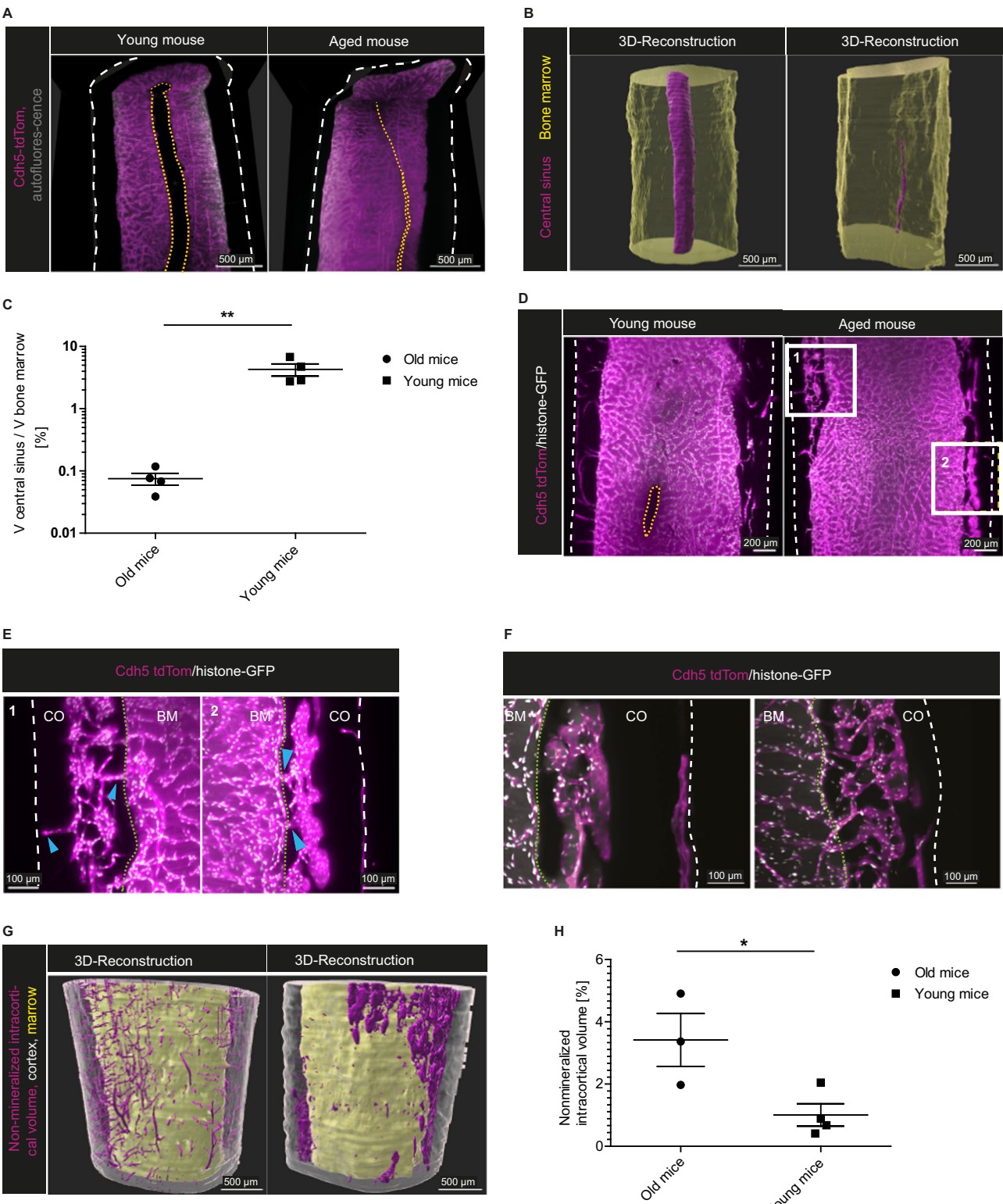

the variability of human annotation of the same data set, providing an intrinsically calibrated quality measure. In this way, a reliable auto-mated segmentation of vascular, stromal and hematopoietic com-partments in the BM has been achieved. Prospectively, even rare cells, such as hematopoietic stem cells, can be phenotypically characterized and quantified, whilst obtaining information about their exact locali-zation within the whole BM. For the same reasons, MarShie is ideally suited for analyzing the development of hematopoietic as well as non-hematopoietic tumors, where niches in the BM play crucial roles.

Our data show that MarShie is compatible with several fluorescent dyes and a wide range of fluorescent proteins. While the red tdTom fluorescence is readily detectable throughout the whole BM, the GFP signal is only weakly detectable deep in the tissue, which is not unex-pected, due to the strong tissue autofluorescence in the green spectral range[47]. Thus, the detection of compact GFP+ structures, such as cell bodies, is possible, but fine sub-cellular structures are no longer dis-cernible. However, by adding ATTO-647N coupled anti-GFP nano-bodies, we enable the detection of even fine cellular projections, and

**Fig. 4 | Age-related vascular changes in the bone marrow. A** xy- and xz-orthogonal slices (50 μm thick) of the femoral diaphysis show a significant volumetric reduction of the central sinus (yellow dashed lines) during aging. Left: young mouse (Cdh5-tdTom reporter mouse, 25 weeks), Right: aged mouse (Cdh5-tdTom reporter mouse, 84 weeks). Cdh5-tdTom signal in magenta. Outer cortical borders are marked by white dashed lines. Images represent four independent experiments. **B** 3D-segmentation from diaphyseal whole mounts, same samples as in (**A**). Bone marrow volume in yellow, central sinus volume in magenta. **C** Proportion of sinus volume to total marrow volume in the diaphysis (analysis 2000 μm in y-axis) comparing young (20–25 weeks) and aged (84–86 weeks) mice. Data of $n = 4$ per age group (mean ± SD) analyzed by two-tailed, unpaired $t$ test, $p = 0.0044$. Source data are provided as a Source Data file. **D** Representative xy-orthogonal slices (50 μm thick) of the diaphysis highlighting the cortical structure. Left: young, Cdh5-tdTom/histone-GFP mouse (20 weeks) and Right: aged mouse (86 weeks). Three and four independent experiments of old and young mice, respectively. Cdh5-tdTom[+] in magenta, histone-GFP[+] in white, bone outlines marked by white dashed lines, central sinus marked by yellow dashed lines. **E** Zoom-ins of areas marked by white rectangles in (**D**), showing intracortical marrow islands. Blue arrows point to vascular connections between those structures and the marrow cavity and periost. Endost marked by green dotted lines, bone outlines marked by white dashed lines in (**E**, **F**), BM bone marrow, CO cortex. **F** Examples of convoluted and branched intracortical vessels in aged mice, different regions from the individual in (**D**) are shown (50 μm thick sections). Cdh5-tdTom signal in magenta, histone-GFP signal in white, autofluorescence in gray. **G** 3D-Segmentation of cortex (gray), non-mineralized cortical volume (magenta) and marrow volume (yellow) in whole-mounts of the samples shown in (**D**). **H** Non-mineralized intracortical volume in the distal diaphysis (analysis 2000 μm in y-axis) comparing young and aged mice. Data of $n = 3$ (old), $n = 4$ (young) mice (mean ± SD) analyzed by two-tailed, unpaired $t$ test, $p = 0.0336$. Source data are provided as a Source Data file.

prove MarShie´s compatibility with staining by antibody-based techniques. Fluorescence signals within the samples remain stable over months, enabling storage, further analysis, or additional staining over extended time periods. Furthermore, it is also possible to perform antibody-based in vivo labeling, as exemplified by staining using AL-647–coupled antibodies directed against CD31, as well as by the staining of myeloid cells using anti-CD169 coupled to eF660. While staining conditions for different markers will have to be optimized separately, our data provide evidence that MarShie enables in vivo staining and subsequent clearing, for stromal as well as hematopoietic cell types.

The possibility to perform whole tissue staining reduces dependence on fluorescent reporter mouse strains in the future, and the combination of various dyes could help to expand the number of parameters that can be investigated within one sample by LSFM. This approach will be crucial to further characterize the heterogeneous microenvironments of the BM[32]. The development of reagents, particular antibodies or fluorescent reporter proteins in the near infrared-wavelength range will provide additional benefits for deep tissue imaging. Along that line, cyclic staining and bleaching approaches, similar to the ones we used in conventional 2D histology[48,49] could be an attractive option.

The image quality achieved by Marshie allows conclusions on the level of subcellular resolution. On the other hand, MarShie enables us to analyze stromal cell subsets which are hard to isolate from the BM, and whose dimensions cannot be determined by means of 2D histology. Due to the ability to visualize cellular structures throughout the marrow at high resolution, MarShie will enable us to characterize stromal networks formed by the delicate extensions, which those cells form. It will provide insights into the understudied heterogeneity of those cells[50,51], and link their phenotypes to certain locations and cellular interactions in the marrow.

Finally, MarShie has potential applications in clearing human bones and addressing questions on how bone and the cells in the BM communicate with their surrounding tissues. Along that line, in preliminary experiments, we could confirm the suitability of MarShie for the preservation and clearing of other tissue types.

Overall, we demonstrate here a protocol that facilitates the analysis of single cells in the marrow tissue of intact whole long bones of mice. MarShie preserves the 3D tissue architecture while resolving even subcellular structures in the deep marrow regions, and therefore holds the potential to resolve long-standing questions in BM biology.

## Methods
### Animals
**Ethical statement.** The research presented in this manuscript complies with all relevant ethical regulations. All experimental procedures involving animals were carried out after approval of the study protocols by the Landesamt für Gesundheit und Soziales Berlin, animal license numbers (G302/17 & G0025-22). Mice were kept under specific pathogen free conditions, maintaining a 12-h light/dark cycle with the ambient temperature set to 22 ± 2 °C and air humidity 55 ± 10% rH, in the animal facilities of the Charité Crossover (CCO) facility and/or the DRFZ. Food and autoclaved water were provided ad libitum and animals were housed in individually ventilated cages (IVCs), containing an enriched environment. Animal experiments were conducted following the 3 R Principles[52]. Mice were handled using tunnels to reduce stress and anxiety.

**Animal procedures.** For this study, we chose mice on a C57/BL6J background. Cdh5-tdTomato/histone-GFP double reporter mice[27], a gift of Prof. Ralf Adams, Münster (Germany) were used for the visualization of endothelial membranes and nuclei. Three female mice, 22–38 weeks old, were used. Adding to that, we crossed these mice with CX₃CR1-GFP[+] animals (The Jackson Laboratory, stock No. 005582) to obtain a CX₃CR1-GFP x Cdh5-tdTomato/histone-GFP genotype for the visualization of endogenous fluorescence of vessel subsets and myeloid cells[27,40]. Mice were bred in the facility of the Charité – Universitätsmedizin Berlin, Germany.

In Prx1-Cre R26-LSL-YFP and Prx1-Cre R26-LSL-tdRFP mice (herein referred to as Prx1-YFP or Prx1-RFP mice), The expression of Cre recombinase under control of the *prrx1* promoter[53] results in Cre recombinase-mediated excision of a *loxp*-flanked STOP codon (LSL), enabling the expression of yellow or red fluorescent protein (YFP/tdRFP), inserted into the ROSA26 locus[54]. As Prx1 is expressed in mesenchymal cells of the developing limb bud, this generates a fate map reporter mouse of stromal cells in bone and BM. Prx1-Cre and R26-LSL-YFP were obtained from The Jackson Laboratory, stock No. 005584 and 006148, respectively. R26-LSL-tdRFP[30] were a gift from Prof. H.J. Fehling, Ulm, Germany. Mice were bred in the facility of the Deutsches Rheuma-Forschungszentrum Berlin, Germany. We used two female Prx1-YFP mice (23 weeks old), and four female Prx1-RFP mice, aged between 17 and 24 weeks.

CD19-Cre R26-LSL-tdRFP mice express Cre recombinase under control of the *cd19* promoter. This is achieved by crossing the CD19-Cre strain (The Jackson Laboratory, stock No. 006785) onto R26-LSL-tdRFP mice[30]. Excision of the LSL results in expression of tdRFP in B lineage cells, as described previously[4]. Three female mice were used at 28 weeks of age. The mice were bred in the facility of the Deutsches Rheuma-Forschungszentrum Berlin, Germany.

The expression of the fluorescent proteins monomeric Kusabira Orange (mKO2) and monomeric Azami Green (mAG) in Fucci (fluorescent ubiquitination-based cell-cycle indicator) mice is linked to the cell-cycle dependent ubiquitination proteins hCdt1 (Fucci 639) and hGeminin (Fucci 474)[31]. Thus, cells in the G1 phase of the cell cycle show nuclear orange fluorescence, while nuclei of cells in the S/G2/M phase fluoresce green. Breeder mice were obtained from Dr. Atsushi Miyawaki, Wako City, Japan, and bred in the mouse facility of the

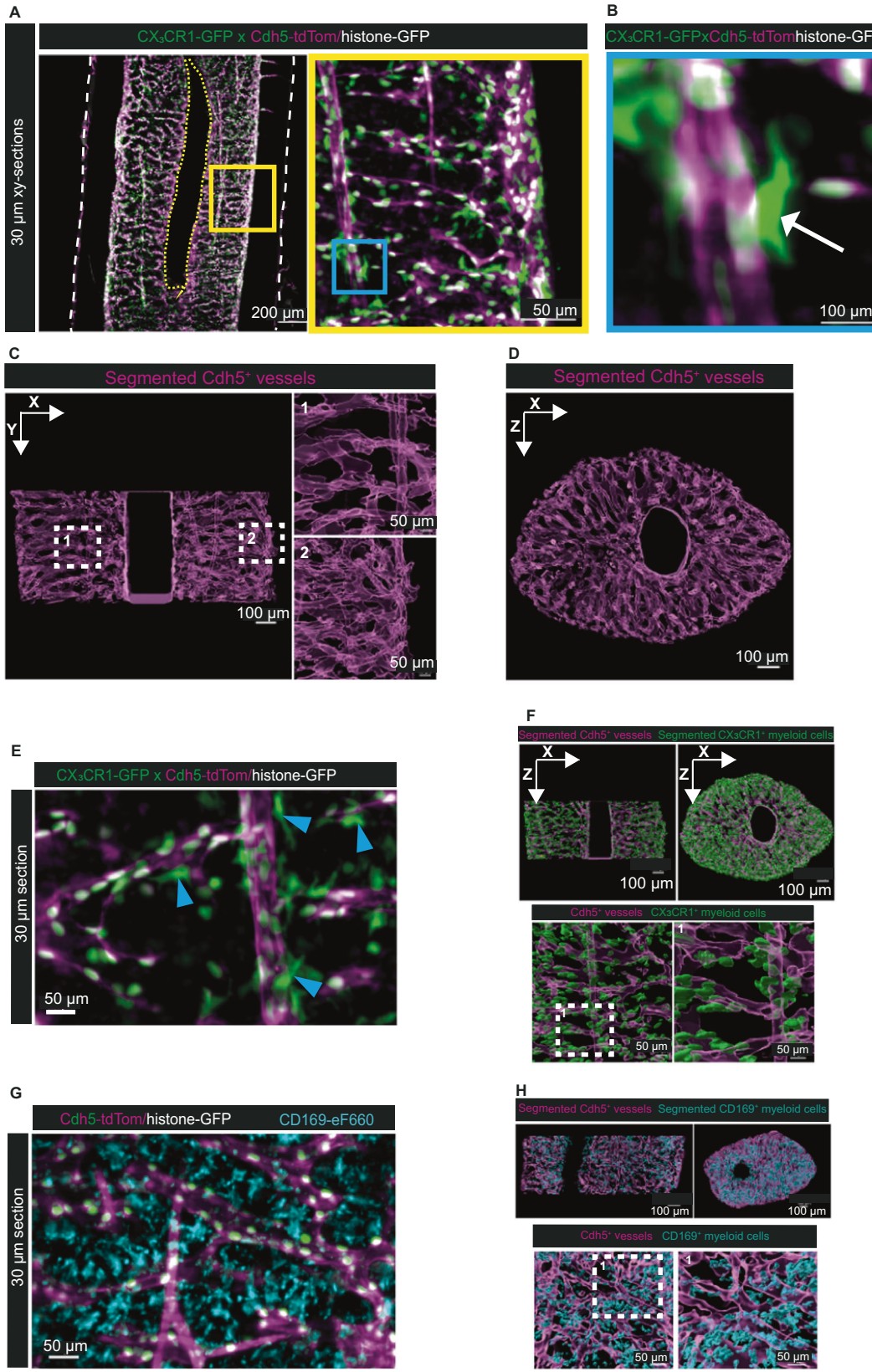

Charité – Universitätsmedizin Berlin, Germany. Regarding the Fucci 639 strain, two females at 16 weeks of age and two males at 35 weeks of age were used. Regarding the Fucci 474 strain, we used four females at 10–13 weeks of age.

Animals carrying a Cre allele under control of the adiponectin (*adipoQ*) promoter (B6;FVB-Tg(Adipoq-cre)1Evdr/J, available from The

Jackson Laboratory, stock no: 010803), which confers Cre recombination in marrow-resident adipocytes, were crossed to the mTmG strain (Gt(ROSA)26Sortm4(ACTB-tdTomato,-EGFP)Luo; The Jackson Laboratory stock No. 007576). The resulting strain expresses membrane-bound Tandem-Tomato (mT), resulting in red fluorescence in all cells before recombination, while Cre-mediated excision results

**Fig. 5 | Machine-learning based analysis of hematopoietic – vascular interactions. A** xy-orthogonal slices (30 μm thick) of a young CX₃CR1⁺-GFP x Cdh5-tdTom/histone-GFP reporter mouse (21 weeks old) show interactions between CX₃CR1⁺ myeloid cells and Cdh5 expressing vessels. Cdh5-tdTom signal in magenta, histone-GFP signal in white, CX₃CR1-GFP signal in green. Image representative for $n = 3$ mice from 3 different experiments. Left: Deep BM diaphysis. Right: Zoom-in of the area within the yellow rectangle highlights CX₃CR1⁺ cells within deep marrow. **B** Zoom-in of the area within the blue rectangle in (**A**) shows a single CX₃CR1⁺ cell (white arrow) close to a vessel. Fine processes extend from the CX₃CR1⁺ cell. **C** xy-orthogonal enlarged slice of segmented Cdh5⁺ vessels (magenta) in the BM. White dashed squares indicate zoom-ins: (1) Segmentation at the deep BM region shows straight arterial-like vessels and transverse sinusoidal vessels, (2) Segmentation at the endosteal region highlights the densely connected vascular network. **D** xz-orthogonal enlarged slice of the Cdh5⁺ segmented vessels demonstrates homogenous segmentation throughout the whole-mount. **E** Examples for the close association of CX₃CR1⁺ cells with Cdh5-tdTom⁺ vessels. Blue arrowheads point at myeloid cells closely associated with vessels. Representative for $n = 3$ mice from 3 different experiments. **F** Upper panels: xy- and xz-orthogonal slices of Cdh5⁺ segmented vessels (magenta) and CX₃CR1⁺ segmented myeloid cells (green) demonstrate reliable segmentation in the deep BM. Lower panels: Enlarged view of 3D segmented CX₃CR1⁺ myeloid cells shows proximity to endothelial cells and absence of CX₃CR1⁺ cells in parenchymal areas distant from vessels (left). Zoom-in of the white dashed square highlights CX₃CR1⁺ myeloid cells in close contact with vessels (right). **G** CD169⁺ myeloid cells are localized throughout the parenchyma. Representative for $n = 3$ mice from 3 different experiments. **H** Upper panels: xy- and xz-orthogonal slices of Cdh5⁺ segmented vessels (magenta) and CD169⁺ segmented myeloid cells (cyan). The signal intensity of antibody-stained myeloid cells allows segmentation in deep BM. Lower panels: Enlarged view of 3D segmented CD169⁺ myeloid cells demonstrates the distribution of those cells throughout the parenchyma (left). Zoom-in of the white dashed square shows CD169⁺ cells not contacting vessels (right).

in a switch to membrane-bound GFP (mG) expression in adipocytes[29]. Four female mice were used at 18–28 weeks of age. This strain was bred in the mouse facility of the German Institute of Human Nutrition (DIfE), Potsdam-Rehbruecke, Germany.

When mice of different ages were compared, mice aged 19 to 25 weeks were considered young, while aged mice were at least 18 months old.

### MarShie tissue clearing protocol

A protocol with step-by-step instructions can be found in the supplemental information. In brief, mice are deeply anesthetized by an intraperitoneal injection of high-dose Ketamine (250 μg/g body weight) and Xylazine (25 μg/g body weight), in order to perform perfusions. Optional intravenous (i.v.) antibody staining through the tail vein is performed 20 min prior to sacrifice. Antibodies used for i.v. staining were 15 μg CD31 conjugated to Alexa Fluor 647 (BioLegend, catalog number 102516, clone: MEC13.3) and 20 μg Anti-Mo CD169 conjugated to eFluor 660 (Invitrogen, catalog number 2211012, clone: SER-4). Once hind leg pedal reflexes are absent, indicative of deep anesthesia, mice are transcardially perfused with ice-cold 1x phosphate buffered saline (1x PBS). In order to do so, we open the abdomen, cut laterally through the ribcage and expose the heart, where a cannula is inserted into the left ventricle and subsequently connected to a pump device (ISMATEC BVP Standard, 8,5 rpm, 5 ml/min). A small incision is made into the right atrium for drainage of blood and perfusion solution. After initial perfusion with 100 ml 1x PBS, we continue to perfuse the animal with 30 ml of ice-cold SHIELD perfusion solution (50% v/v SHIELD Epoxy, 25% v/v SHIELD Buffer solution (life canvas) 4% w/v paraformaldehyde (electron microscopy sciences), 21% v/v distilled water).

Once fixed, the femurs and tibias are extracted and carefully cleaned of residual muscle and surrounding connective tissue. Then, we postfix the bones in 15 ml Falcon tubes filled with SHIELD Perfusion solution for 2 days at 4 °C, on a rotating device (MACSmix, Miltenyi Biotec, 20 rpm). Next, we incubate the samples for one day in SHIELD OFF solution at 4 °C, followed by SHIELD ON solution at 37 °C the next day. Following these steps, our samples are sufficiently fixed and polyepoxidized.

For decalcification of the mineralized bone tissue, we treat the samples with 20% w/v EDTA (Carl Roth) at pH 8 in 1x PBS solution for 5 days, at 4 °C, in the rotating device. We exchange the EDTA solution daily. Next, we use the SmartClear II Pro device (Life Canvas) to actively de-lipidate our samples at 42° for 1–3 days (1500 mA, 90 V).

To decolorize our samples for optimal transparency, bones are treated in a mixture of 10% w/v CHAPS (3-[(3-Cholamidopropyl)-dimethylammonio]-1-propansulfonat) (Carl Roth) and 25% w/v Quadrol (N,N,N′,N′-Tetrakis(2-Hydroxypropyl)ethylendiamin) (Sigma-Aldrich) in 1x PBS at room temperature, rotating at 20 rpm for 2 days.

Where indicated, we boost the GFP and YFP signals with Atto647N conjugated to an anti-GFP nanobody (nanobooster, ChromoTek GmbH, gba647N-100) using a protocol which we adapted from the mild vDISCO staining protocol (Cai et al., 2019). To boost the whole mounts, we block the bone samples for 1 day in 5,5 ml blocking buffer containing 1.5% goat serum (Thermo Fisher) and 0.5% Triton X-100 (Sigma-Aldrich) in 1x PBS at room temperature. We incubate for 5 days in fresh blocking buffer with a total volume of 5,5 ml. To boost the CX₃CR1-GFP signal, we add 8 μg of the nanobooster, 12 μg are used to boost the Prx1-YFP signal. For CX₃CR1-GFP reporter mice, we boost both contralateral femurs within the 5,5 ml staining solution, whereas we only boost one femur at a time for Prx1-YFP reporter mice. Finally, we wash our samples 3x for 1 h in the same solution (1,5% goat serum, 0,5% Triton X-100 in 1x PBS) followed by 3x 1 h of 1x PBS washes. All steps are performed at room temperature, in the dark and under constant rotation.

For refractive index matching, we gradually submerge the samples in ascending concentrations of Easy Index (life canvas), starting with a mixture of Easy Index 1.47 and 1x PBS (50:50) for one day, and changing into the final solution of pure Easy Index 1.52 the following day. We proceed to capture microscopic images of the samples the day after or store them for several weeks until imaging.

### Additional tissue clearing protocols tested

**ECI.** Following the ECI clearing protocol[20] in brief, we transcardially perfuse the mice with 4% PFA in 1x PBS and postfix the femur samples in PBS overnight at 4 °C. Then we incubate the samples in an incrementing EtOH concentration, starting form 50% in PBS, adjusted for a pH of 9 for 12 h, 70% EtOH 12 h and 100 % EtOH for 24 h. All steps are carried out at 4 °C. To match the RI, we immerse the samples in 100% ethyl cinnamate (ECI; Sigma-Aldrich).

**FDISCO.** Following the FDISCO protocol[21] we perfuse the animals with 4% PFA in 1x PBS and postfix the samples in the perfusion solution overnight at 4 °C. Using a shaker, we decalcify the samples in a 0.1 M EDTA-2Na solution for three days at 37 °C, changing EDTA daily. Following decalcification, we immerse our samples in an ascending gradient of tetrahydrofluoran (THF), pH 9 adjusted by the addition of triethylamin in H₂O at 4 °C. We start with 50% THF for 3 h, and progress to 70% THF 3 h, 80%THF 3 h and finally 100% THF overnight. After leaving the samples in fresh 100% THF for another 3 h, we immerse the bones in pure dibenzylether (DBE).

**PEGASOS.** Following the PEGASOS protocol[13], after initial animal perfusion with 4% PFA in 1x PBS we postfix our samples in the perfusion solution overnight at 4 °C. The next day, we place the bones in 0.5 M EDTA (20%) at 37 °C whilst rotating. We exchange the EDTA solution daily for all 5 days of decalcification. After a subsequent washing step

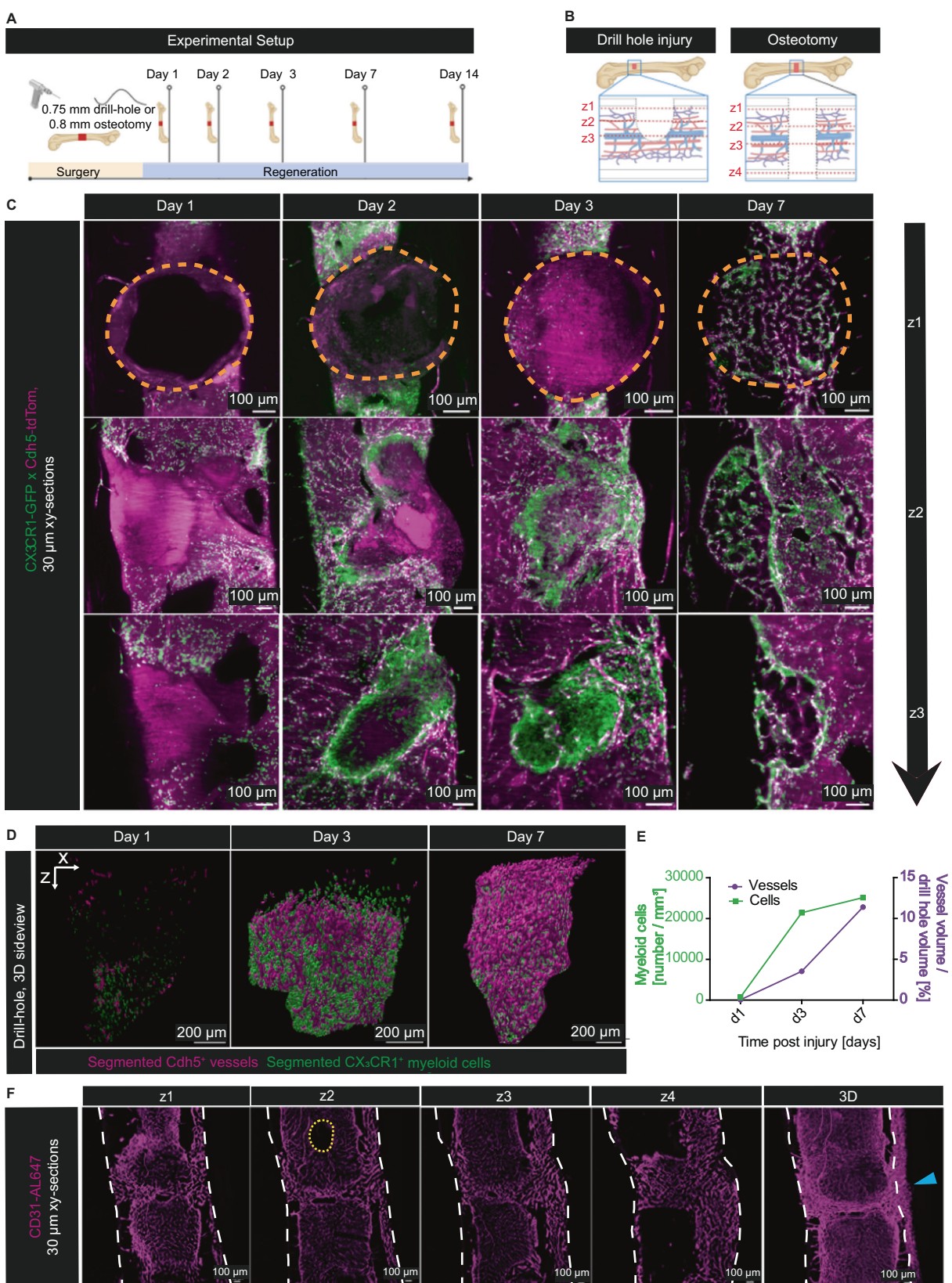

with H₂O, we immerse the samples into a 25% v/v mixture of Quadrol in H₂O for a total of two days. For decolorization, we place the samples into a 5% Ammonium solution in H₂O for 6 h and dehydrate the bones in 30% tert-butanol in H₂O for 4 h, 50% tert-butanol for 6 h and 70% tert-butanol for 24 h. Additionally we add 3% v/v Quadrol to the tert-butanol gradients. For the next two days, we place the samples in a mixture of 75% tert-butanol + 22% PEGMMA + 3% Quadrol. To finish the protocol, we incubate the bones in a BB-PEG reagent of 75% BB (benzyl benzoate) + 22% PEGMMA + 3% Quadrol for 24 h before imaging.

**Bone CLARITY**. Following the Bone CLARITY protocol[12], after perfusion and postfixation analogous to the above-mentioned protocols, we

**Fig. 6 | Analysis of bone regeneration models in 3D using MarShie. A** Schematic of the experimental setup for analyzing bone regeneration after either drill hole injury or osteotomy. Illustration created with BioRender.com. **B** Schematic of the drill hole injury and the osteotomy. Dashed red lines indicate the extent (depth) of each injury shown in (**C**) and (**F**) Illustration created with BioRender.com. **C** Representative xy-orthogonal slices (30 μm thick) at day 1, 2, 3 and 7 after drill hole injury ($n = 5$ per time point, analyzed in 3 independent experiments). Three slices throughout the z-depth of the drill hole injury are shown per time point, as illustrated in (**B**). Cdh5-tdTom in magenta, $CX_3CR1$-GFP in green, histone-GFP in white. Drill hole areas are marked by dashed orange lines. **D** 3D-Segmentation of the Cdh5-tdTom$^+$ vascular system (in magenta) and the $CX_3CR1$-GFP$^+$ myeloid cells (in green) within the drill hole at day 1, 3 and 7 after injury. **E** Quantification of $CX_3CR1^+$-GFP$^+$ myeloid cell numbers per mm$^3$ and Cdh5-TdTom$^+$ endothelia cells (vessel volume/drill hole volume) over time. Source data are provided as a Source Data file. **F** Representative xy-orthogonal slices (30 μm thick) at day 14 after osteotomy show abundance of CD31$^+$ vessels throughout the fracture gap ($n = 5$, analyzed in three different experiments). Four slices throughout the z-depth of the osteotomy gap are shown as indicated in (**B**). Blue arrowhead points at massive medial periosteal thickening and sprouting into the osteotomy gap. CD31-Al647$^+$ in magenta (25 μm background subtracted); upper border of drill hole injury is outlined by dashed orange lines, outer cortical borders are marked by dashed white lines, central sinus is outlined by dashed yellow lines.

place the samples in 10% EDTA, pH8 at 4 °C for a total of 14 days, with daily solution exchanges whilst rotating. We perform tissue hydrogel embedding in 4% acrylamid, with 0,25% thermoinitiator VA-044 in 1x PBS, overnight at 4 °C. After a short degassing step of 5 min, we polymerize the hydrogel for 3 h at 37 °C in a water bath. Once the hydrogel stabilizes the tissue architecture, we de-lipidate the samples in 8% sodium dodecyl sulfate (SDS) in 1x PBS for five days. Thereafter, we wash the tissue in 1x PBS for 2 days to fully remove any SDS residues. After a decolorization step of 25% wt/v Quadrol in 1x PBS for 2 days we again wash the samples in 1x PBS for another day. Following the protocol, we immerse the samples in an ascending gradient of refractive index matching solution (RIMS) with daily incubation in RIMS RI 1.38, RI 1.43 and final solution with a RI of 1.47.

## Drill hole injury and osteotomy models

Mice are randomly subjected to either a drill hole surgery or an osteotomy of the distal diaphysis of the right femur. All animals undergoing drill hole surgeries or osteotomies in this study were female and aged between 19 and 25 weeks. Mice are anesthetized with 1.5–2.0% isoflurane via an inhalation mask. After applying buprenorphine (0,03 mg/kg body weight) as analgetic agent and enrofloxacin (10 mg/kg body weight) as antibiotic, both s.c., the fur covering the surgical surface is shaved, and the underlying, naked skin disinfected with Braunol (B|Braun). Following that, a lateral incision of ~1.5 cm is performed between the right knee and hip joint, parallel to the femur, under microscopic control. The underlying muscle tissue is dissected and retracted along the fascia. Using a ring forceps, the femur is fixed in position. For drill-hole injuries, a 0.75 mm-wide hole in the distal third of the diaphysis is drilled through the cortex, into the marrow using an electric precision drill (Proxxon GmbH) until nearly reaching the contralateral cortical boundary. Careful hemostatic control is applied by gentle pressure using sterile swabs. For osteotomy defects, we place an external fixator (RISystem) parallel to the right femur. Four serial pins, drilled to through the cortex hold the system in place. We procede to create the 0.8 mm osteotomy gap with a gigli wire saw (RISystem). Wounds are sewed with an absorbable surgical thread (Surgicryl rapid). The anesthetized animals are kept on a temperature-controlled surgery plate during the whole procedure. Antibiotic prophylaxis with enrofloxacin (10 mg/kg body weight) is administered twice daily for the following three days via s.c. application. Tramadol (50 mg/l) is added to the drinking water for postoperative pain relief, in addition to glucose (0.06 mg/ml $H_2O$) to increase the acceptance by the animals because of the bitter taste of the medication.

## Sample mounting and light-sheet fluorescence microscopy image acquisition

We carefully glue the femoral bones at both epiphyses on a lancet (B|Braun 1 × 2000 Solofix) and immerse the samples in a 5 × 10 × 45 mm fluorescence cuvette, optical glass (msscientific Chromatographie-Handel GmbH) mounted in Easy Index RI 1.52. The small cuvette containing the sample is glued onto the sample holder and subsequently immersed in the bigger imaging cuvette (LaVision Biotec – a Milteny company, Bielefeld, Germany) filled with Cargille Immersion Oil Type LDF, RI 1.515.

To acquire microscopic image stacks we use an Ultramicroscope II (LaVision Biotec – a Milteny company, Bielefeld, Germany) coupled to an Olympus MVX10 zoom body, providing a zoom ratio from 0.63x - 6.3x. We use a 2x dipping objective (Olympus MVPLAPO2XC/0.5 NA fitted with a 5.7 mm working distance dipping cap, resulting in a total magnification ranging from 1.36x to 13.56x). The Ultramicroscope II, featuring an axial resolution of about 4 μm, is equipped with an Andor Neo sCMOS Camera with a pixel size of 6.5 × 6.5 μm² and a LaVision BioTec Laser Module. We used the following filter sets for image acquisition: ex 488 nm, em 520/50 nm for autofluorescence, GFP and YFP signal; ex 561 nm, em 620/60 nm for tdTomato signal; ex 640 nm, em 680/30 nm for anti-GFP-Atto647N signal and CD31-Alexa 647.

Adjusting the zoom body, we use total optical zoom factors between 1.25x and 4x and a z-step-size between 3 and 5 μm. Larger tile scans are acquired using a 10% overlap along the longitudinal y-axis of the bone. We adjust the laser power depending on the intensity of the signal, in order to avoid saturation. In paralleling image acquisitions, laser powers are kept in a comparable range and only fine adjusted within a few percent of laser power. Exposure time is set at around 300 ms. We use the dynamic focus settings whenever deemed appropriate. The light sheet width is adjusted between 80 and 90% for a homogenous illumination of the field of view.

## Image processing and analysis

We separately acquire 16-bit grayscale TIFF images for each channel with the ImSpector Pro software (LaVision Biotec). Tiff stacks are converted (Imaris File Converter, Bitplane AG) into Imaris files (.ims) and stitched with Imaris Stitcher. We use Imaris x64 software (version 9.9.1) for 2D and 3D image visualization, snapshot creation and movie generation using the snapshot and animation tools. To illustrate the BM samples we use the 3D view and optically slice the tissue using the ortho tool from different angles and depths. Whenever appropriate, we use clipping planes to better illustrate structures of interest within the bone.

For visualization of some individual sections in the figures, using the image processing functions in Imaris, we perform a background subtraction algorithm as mentioned in the figure legends.

## Directional FFT algorithm for stripe artifact suppression in light-sheet images

In our microscope setup, laser illumination is simultaneously achieved along six co-planar directions for homogeneous excitation, with the illumination plane perpendicular to the detection axis. The six laser beam paths are arranged in two groups of three, oriented in 180° to each other, with an angle of 12° between the beam paths within each group. This results into a quasi-periodic pattern of stripe artifacts in the detected signal, related to the laser illumination pattern. The extent of the stripe artifacts depends on the level of the detected signal: the lower the SNR, the stronger the artifacts. For generating the appropriate filter to exclude frequencies associated with stripe arti-

facts and preserve structure integrity in the image, we first compared Fourier transformed stripy images with the geometry of the illumination pattern of the microscope. In the Fourier transformed images, we found the directions associated with stripe artifact pattern to be similar to the three directions (−12°, 0°, 12°) of the illumination pattern. However, presumably due to loss of resolution by diffraction and beam divergence of the light sheet, the striped pattern is described by arcs of circle of ≈3°, rather than sharp lines, along each direction. This leads to a slight deviation in the orientation of the central lines in each of these arcs of circle, i.e., (−10°, 0°, 10°), as compared to the orientation of the illumination pattern.

To generate the directional filter in Fourier space, sharp lines along the central directions of the striped pattern (−10°, 0°, 10°) are defined and given the value 1, within a zero-filled FFT image of the same size as the original image. Next, we blur these sharp lines with Gaussian functions, increasing the width of the Gaussian linearly along each direction, from the center to the edge of the FFT image. Thus, we account for the shape of the light sheet, i.e., light sheet width and divergence. Finally, we apply a band pass filter defined as the difference of Gaussians (doG), to preserve lower and higher frequencies in the filter mask, i.e., to preserve structures either larger or smaller than the period of the striped pattern, which are less affected by the stripe artifacts. Both lower and upper cutoff frequencies are estimated in the FFT representation of the original images by Gaussian fitting of the frequency distribution along each of the three directions of the striped pattern. The product of the Fourier-transformed original image and of the inverted de-striping filter is inversely Fourier transformed and, as the stripe artifacts are suppressed, further used for image segmentation.

### Quantification of central sinus volume in young and aged mice

For comparison of volumetric changes in the central sinus vasculature, we compare young mice ($n = 4$), aged 20 to 25 weeks to old mice ($n = 4$), aged 84–86 weeks.

We restrict our analysis to the mid-diaphyseal bone volume. In order to optically isolate our volume of interest, we crop our image stacks to a uniform height (y-axis) of 2000 μm for further analysis. From there, we use the Imaris surface segmentation with the manual-surface creation tool to construct a three-dimensional representation of the central sinus, as well as of the whole BM in the same portion of the mid-diaphyseal volume. Using the semi-automatic threshold tool isoline, we mark the surface borders of the central sinus and the BM separately in xz direction for every 50 μm section, spanning the entire length of our volume. Then we create the surfaces using the preserve features algorithm in the software for a representative reconstruction. Subsequently, we obtain the surface volumes through the Imaris statistics tab. We divide the central sinus volume by the whole BM volume, and multiply the result by 100 to obtain the relative percentage of central sinus volume within the total BM, in young and aged individuals.

### Volumetric quantification of intracortical marrow in young and aged mice

For comparison of volumetric changes in intracortical deposition of BM in the distal diaphysis, we compared young mice ($n = 4$), aged 20–25 weeks to aged mice ($n = 3$), aged 82–86 weeks.

For quantification of age-related marrow deposition in the distal diaphyseal cortex, we first crop our image stack to a uniform length of 2000 μm. Then, following the clearly recognizable outer boundaries of the cortex and inner endosteal cortex lining, we optically isolate the cortical volume by generating Imaris surfaces of the entire bone, as well as of the BM compartment. Whenever possible, we use the semi-automatic isoline tool for intensity-based annotation in a stepwise manner, labeling the boundaries of the surfaces at least every 50 μm in xz direction. When a straightforward intensity-based annotation does

not result in the correct segmentation, we instead use the manual drawing tool to follow the recognizable outlines of the surfaces. By masking our image channels, we are left with a signal that is only contained inside the boundaries of the cortical bone. Intensity signals outside the whole bone surface and inside the BM are set to 0. From there on, we use the Imaris LABKIT interface[35] as a machine-learning tool for pixelwise semantic segmentation, to label the intracortical marrow islands in aged mice, and the transcortical vessels in young mice, for subsequent volumetric quantification. Based on sparse user annotations, a random forest classifier is trained on the user label input. This allows for semantic segmentation based on the following filters: [gaussian blur, difference of gaussians, gaussian gradient magnitude, laplacian of gaussian, hessian eigenvalues] for sigmas 1, 2, 4, 8 in three dimensions. In the closing filter steps, we discard objects smaller than 100 μm³ (corresponding to a sphere with a diameter of $d \approx 5{,}76$ μm) to remove the remaining noise.

Using the formula

$$V[\%]\text{intracortical vessel volume} = \frac{\text{Nonmineralized intracortical volume}}{V(\text{total bone}) - V(\text{bone marow})} \cdot 100$$

(1)

we obtain the relative contribution of intracortical vessels to the total volume of the distal diaphyseal cortex.

### Evaluation and quantification of bone regeneration in drill hole injuries

For visualization of the time course of fracture healing following drill hole injuries, we select images of upper, middle and lower third (in z dimension) of the respective defects, in order to account for the heterogeneity in injury patterns. For every time point, we choose a representative image from one of at least three individuals.

For segmentations of the drill hole defect volumes in Imaris, we manually track the boundaries of the drill hole defect, at least every 50 μm, plane by plane in xz orientation. We mark the interface between organized unaffected and unorganized injured tissue, based on the autofluorescence of the tissue, and convert it into a surface model.

Using machine learning in LABKIT[35], we first segment the Cdh5-tdTomato⁺ blood vessels. Next, in order to exclude the GFP⁺ endothelial cell nuclei, for the anti-GFP boosted channel we mask the inside of the blood vessel surface and set the pixel values to zero. By doing so, we are left with only the extravascular, GFP boosted signal from the CX₃CR1⁺ myeloid cells, enabling us to segment these cells using LABKIT in the following step. We further filter the resulting segmented objects by size, with a lower threshold of 180 μm³, to include only CX₃CR1⁺ cells. It is essential to enable the split touching object algorithm in Imaris, with a seed point diameter of 11 μm, to separate the cell bodies from one another as they commonly appear in close contact and display a highly branched, dendritic phenotype.

### Fluorescence signal quantification

We quantify the signal to noise ratio (SNR) for the GFP, tdTom and the Atto-647N signal by

$$\text{SNR} = \frac{(\text{Mean Foreground} - \text{Mean Background})}{\text{Standard deviation of Background}}$$

(2)

For the quantification, we choose representative ROIs of $500 \times 500$ μm from three consecutive z-slices in endosteal and deep marrow regions of the diaphysis. Data from three different animals are included in the analysis. We segment CX₃CR1-GFP⁺ myeloid cells plus histone-linked GFP⁺ endothelial nuclei, as well as the nanoboosted anti-GFP-Atto-647N signal and Cdh5-tdTomato⁺ vessels with LABKIT and calculate the mean signal intensities of those three signals. For quantification of mean background, for each channel we select 20 equally

sized circular regions inside the marrow without notable signal. Per definition, a SNR value of 1 indicates that the signal equals the background noise and, thus, cannot be detected. Based on our experience and published data, SNR values of 4 are sufficient for image segmentation[55].

We calculate the signal to background ratio (SBR) by

$$SBR = \frac{\text{Mean Foreground (signal)}}{\text{Mean Background}} \quad (3)$$

to quantify the SBR preservation by means of the Cdh5-tdTom$^+$ signal over time. Again, we choose $500 \times 500\,\mu m$ representative ROIs from deep marrow diaphyseal regions of single image planes of three to four different animals per time point. To quantify the mean background, we choose 15 equally sized circular regions in Fiji (ImageJ)[56] inside the marrow without notable vessel signal (Fig. 2H and Supplementary Fig. 2B). We segment the Cdh5-tdTom$^+$ vessels with LABKIT to obtain the mean foreground pixel intensity (Fig. 2H and Supplementary Fig. 2C).

To visualize intensity differences across signal structures of interest, we generate line plots traversing the Cdh5-TdTom$^+$ (Fig. 2D) and anti-GFP Atto647$^+$ (Fig. 2F) signal and plot the signal intensity normalized to surrounding background. To do so, we quantify background fluorescence by a line plot across the local background using the plot profile function in Fiji and divide signal intensities by the mean background fluorescence (Supplementary Fig. 2D).

### Segmentation and quantification of Cdh5-tdTomato$^+$ vessels, the histone-linked GFP of endothelial cells and CX$_3$CR1-GFP$^+$ myeloid cells

We restrict our analysis to a whole-mount mid-diaphyseal BM volume. We crop our image stack (CX$_3$CR1-GFP x Cdh5 tdTom/histone-GFP mouse, 21-weeks-old) of the femoral bone to a height (y-axis) of $500\,\mu m$ for analysis. In Imaris, whenever appropriate we use the semi-automatic threshold tool isoline to segment the volume of the BM. For that, we mark the endosteal borders, slicing plane by plane for every $25\,\mu m$ in xz orientation. For the distinction of the different sub-volumes, we always use the 488 nm excitation channel, clearly showing a difference in the autofluorescence signal between cortex and the BM volume. To segment the intracortical volume, we use the manual drawing tool for every $50\,\mu m$ plane across the whole volume in xz orientation. We exclude the intracortical volume from our analysis by masking the respective fluorescence signal. We segment the central venous sinus with the semi-automatic threshold tool isoline by marking its borders, clearly distinguishable from the rest of the marrow by evident loss of autofluorescence, for single planes every $25\,\mu m$ across the stack. We include extensions of large sinusoidal vessels, which converge into the main sinus in our segmentation. After segmentation of the central sinus, we mask its Cdh5-tdTom and anti-GFP-Atto647N signal by subtracting the fluorescence signal from the signal within the BM volume.

We segment the vascular network of the BM volume on basis of the Cdh5-tdTom signal. We first pre-process the raw images to suppress stripe artifacts appearing in the marrow regions of this channel (Fig. 3F and Supplementary Fig. 3C).

Thereafter, using LABKIT, we perform semantic segmentation by assigning pixels of a subset of the image data to background (non-Cdh5-tdTom) and foreground classes (Cdh5-tdTom) at different z-slices across the volume. We train the algorithm to detect straight, smaller arteries as well as larger sinusoidal vessels including their inner volumes at different z-steps and regions. We use the same parameters as mentioned before for the random forest-based pixel classification algorithm. We discard individual volumes smaller than $94\,\mu m^3$ to remove the remaining unspecific signal. We merge the volumes of

central sinus and surrounding BM vasculature, to generate a full representation of the vessel network in Imaris.

In the next step, we train on histone-linked GFP expression of endothelial nuclei based on the raw data of the anti-GFP-Atto647N nanoboosted signal using the 640 nm excitation channel. To separate histone-linked GFP from CX$_3$CR1-GFP expression, we set voxel intensities outside the vessel surfaces to a value of 0, so that only the GFP$^+$ endothelial nuclei remain. We train the random forest-based pixel classification algorithm on a subset of image data across the whole-mount by manually annotating histone-linked GFP signal of endothelial nuclei and background. Next, we use the surface tool in Imaris to generate the endothelial nuclei reconstruction. We separate touching objects by applying splitting touching objects using a seed point diameter of $7\,\mu m$ for nuclei. We discard individual objects smaller than $33\,\mu m^3$, which due to their small size do not represent endothelial nuclei, but rather unspecific signals. Similarly, we train on endothelial nuclei of the central sinus by expression of anti-GFP-Atto-647N signal and add their number to the BM vessel volume for our analysis.

Lastly, we segment the CX$_3$CR1-GFP Atto-647N boosted or CD169-eF660 signal using the raw data of the 640 nm channel. To exclude the GFP signal stemming from endothelial nuclei, we mask the voxel intensity inside the vessel surface, so that only CX$_3$CR1-GFP$^+$ myeloid cells outside the vasculature remain. Again, we use the surface tool in Imaris for 3D reconstruction. We apply the random forest-based pixel classification in LABKIT and train on the GFP$^+$ signal of the CX$_3$CR1 myeloid cells where we label the cell bodies as well as their fine processes. Subsequently, we split touching objects with a seed point diameter of $11\,\mu m$ for CX$_3$CR1$^+$ myeloid cells, and exclude objects smaller than $170\,\mu m^3$, which we reason are too small to represent myeloid cells.

### Validation of the segmentation algorithm

To assess the segmentation accuracy of our algorithm we used Intersection over Union (IoU alternatively referred to as Jaccard Index or Tanimoto Coefficient)[46,57] for an overlap-based metric evaluation of the algorithm. To do so, annotations were created by six trained raters, which were then compared to the output of our segmentation algorithm. Here, we chose two representative 3 dimensional image subsets, each measuring $200 \times 200 \times 30\,\mu m$ of Cdh5$^+$ vasculature in regions of endosteal and deep marrow, to account for different volumes.

To create the individual expert annotations we used the web based, open source annotation tool webknossos (https://weblium.webknossos.org) to manually define the vasculature versus the background. The resulting output of binary masks where then compared to the algorithm segmentation previously created in Imaris with LABKIT. In FIJI, we first created an intersection (AND) and a union (OR) image from the input segmentations and subsequently measured their respective integrated density and divided the resulting values. By doing so, for every individual slide and BM region we generated an IoU value ranging from 0 (no overlap) to 1 (perfect overlap). To finish, we averaged the IoU values from every slide to create a distinct mean IoU for the endosteal and deep marrow volume for all the trained raters, compared to our segmentation algorithm result. Following the same approach in FIJI, we also compared the interrater variability of the different manual annotations.

### Acquisition of transmission spectra in femurs

We recorded the transmission spectra in three femurs (C57BL/6J mice, 11 to 16 weeks old), at the different stages of the MarShie clearing process. For that, we used an absorption spectrometer with a halogen lamp as visible light source. The cuvette containing the femurs is placed on a translation stage within the beam-path, for optimal alignment of the light focus into the femur diaphysis. Surrounding medium in the cuvette is either PBS or Easy Index. The transmitted light is analyzed by a spectral detector (Ocean Insight, SR-4VN500-25), with

1.5 nm wavelength resolution. We calibrated the wavelength-dependent sensitivity of the detector by measuring the transmission at 485, 522, 632.8 and 656 nm, respectively, within the relevant spectral range for light-sheet microscopy. In this way, we exclude any bias of the detector sensitivity on the transmission spectra. Prior to each measurement, we record the transmission spectrum of the cuvette containing only 1x PBS or Easy Index, as baseline spectrum. Subsequently, we immerse the femur in the cuvette, in the same medium, and, again acquire the transmission spectrum. We calculate the transmission spectra of the tissue as the difference between the measured transmission spectra of the cuvette containing the sample and the baseline spectrum. Both the detector background and the spectrum of the halogen lamp are determined as averages of three independent measurements, prior to each measurement. For each sample and baseline spectrum, we subtract the corresponding background. Further, we normalize the resulting spectra by dividing them by the spectrum of the halogen lamp. Finally, the spectra are smoothed to remove noise artefacts. The normalized transmission for each of excitation wavelength of the light-sheet microscope is the signal acquired only from the tissue, summed up over a range of 20 nm, centered at the respective wavelength.

### Setup for two-photon microscopy of murine femurs cleared using MarShie

Two-photon microscopic experiments were performed as previously described, using a specialized laser-scanning microscope based on a commercial scan head (TriMScope II, LaVision BioTec – a Milteny company, Bielefeld, Germany)[58]. A near-infrared laser (Ti:Sa, tunable in the range 690–1080 nm, 80 MHz, pulse width 140 fs, Chameleon Ultra II, Coherent, Glasgow, UK) and an infrared laser (OPO, tunable in the range 1050 nm–1350 nm, 80 MHz, pulse width, 200 fs, APE, Berlin, Germany) were used as excitation sources. The Ti:Sa and OPO beams, both linearly polarized, were combined in the scan head using a dichroic mirror. A water-immersion objective lens (20×, NA 1.05, Zeiss AG, Jena, Germany) was used to focus both Ti:Sa and OPO beams into the sample. The laser power was controlled by combinations of $\lambda/2$ wave-plates and polarizers. TdTomato fluorescence and SHG signals were generated using 1100 nm fs-pulsed radiation of the OPO and collected in the backward epi-direction using dichroic mirror (775, Chroma) and directed to PMT detectors (H7422, Hamamatsu, Hamamatsu City, Japan). PMTs were assembled in a detection system with different optical channels, i.e., $466 \pm 20$ nm, $525 \pm 25$ nm, $593 \pm 20$ nm, and $655 \pm 20$ nm, where every channel was determined by individual fluorescence filter and a set of dichroic mirrors. To unequivocally separate SHG and tdTom fluorescence, we used our previously published spectral unmixing approach SIMI[59]. We acquired 3D images of $500 \times 500$ μm$^2$ lateral dimension (digital resolution of $1041 \times 1041$ pixel) and 400 μm depth, with axial step-size of 2 μm.

### Statistics and reproducibility

Statistical analysis is carried out using Graph Pad Prism 5 and Origin 9.0. For the comparison of unpaired groups, we used two-tailed, unpaired Student's t-tests. The p-value was considered statistically significant when <0.05.

Sample sizes were chosen based on preliminary findings and pilot experiments. No data were excluded from the analysis. We conducted multiple independent experiments for all conditions tested, details are given in the figure legends. All animals and samples were randomly subjected to the respective procedures. As the main purpose of this study was to establish a method, blinding was not performed, except for the data shown in Fig. 4C, H.

### Reporting summary

Further information on research design is available in the Nature Portfolio Reporting Summary linked to this article.

## Data availability

The raw imaging data are available under restricted access because of their large size, access can be obtained upon request to the corresponding author. Requests will be fulfilled within 2 weeks. Source data are provided with this paper.

## Code availability

The code for the de-striping algorithm[37] is available on the Zenodo repository under doi 10.5281/zenodo.10374405 and as Supplementary Software 1 to this manuscript.

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

## Acknowledgements

The authors are indebted to Robert Günther and Ralf Uecker for their valuable technical support. We thank the trained raters (Sandy Kroh, Johannes Hankeln and Clara Hankeln) for their support in validating the segmentation. This study was supported by funding from the Deutsche Forschungsgemeinschaft (DFG), SPP1937 (HA5354/8-2), HA5354/10-1, HA5354/11-1 and DFG HA5354/12-1, all to A.E.H., SFB 1444, DFG project ID 427826188, P14 (to A.E.H and R.A.N.), P11 (to A.T.) and P12 (to T.J.S.). R.A.N. was supported by DFG NI-1167/5-1 and NI-1167/7-1. A.E.H and R.A.N were supported by a grant from the Einstein Stiftung Berlin (A-2019-559) and by DFG FOR 5560 (DFG project ID 505372148), HA5354/13-1 and NI1167/9-1. T.F.M. was supported by the Sonnenfeld Stiftung and a stipend from the Charité – Universitätsmedizin Berlin. A.T.L. was supported by Studienstiftung des Deutschen Volkes. T.J.S. was supported by grants within the German Center for Diabetes Research (DZD) funded by the German Ministry of Education and Research (BMBF) and the State of Brandenburg (DZD grant IDs 82DZD00302, 82DZD0042G and 82DZD03C3G), and a grant from the Leibniz Association (ID K398/2021).

## Author contributions

A.E.H. and R.N. conceptualized the study. T.F.M., A.T.L, J.E., A.R., A.W. and W.L. performed experiments. R.K. developed de-striping

algorithms, L.K and G.E. provided expertize for development of the algorithms. T.F.M., A.T.L., J.E., R.K. and R.N. analyzed the data. A.K., A.T. and T.J.S. provided materials. T.F.M., A.L., R.N. and A.E.H. interpreted the results and wrote the manuscript. All authors reviewed the manuscript.

## Funding

## Competing interests

L.K and G.E. are employed by Miltenyi Biotec GmbH. The other authors declare no competing interests.
