## [Peer Review File · Nature Communications]

MarShie: a clearing protocol for 3D-analysis of single cells throughout the bone marrow at subcellular resolutionReviewers' comments:

Reviewer #1 (Remarks to the Author)

This manuscript by Mertens et al. has reported the optimization of an optical clearing method for the 3D assessment of cellular structures in the bone marrow of mouse femurs. In the past decade, many studies have reported the development of 3D imaging strategies for intact organs of various species, including the "hard" tissues such as femurs and the skull. In light of this developing trend in the research field, the overall novelty of this study is not strong. While this work was relatively well pursued and may become publishable, several issues must be addressed.

- (1) The procedure of optical clearing reported in this study is a combination of several established protocols. Whether such incremental improvements would make the current method superior to others is questionable.
- (2) Related to (1), this study was primarily focused on vascular structures and their pathophysiological changes in the bone marrow. Whether those research purposes could be achieved with other established 3D imaging methods or even conventional 2D immunostaining appears unclear.
- (3) The 3D imaging results in this study were mostly achieved via the fluorescence reporter lines, except for the whole-tissue immunostaining of anti-GFP nanobody. Such strong dependence on transgenic mouse lines limits the general utilization of the method.
- (4) The manuscript boasted the implementation of machine-learning-facilitated segmentation of images. However, there needs to be solid validation of the accuracy of the algorithm.

Reviewer #2 (Remarks to the Author)

The authors are to be commended for their interesting study.

However, I have some comments that I feel should be addressed.

1. One of the main study aims was to demonstrate that MarShie was superior to previous clearing protocols. It is stated that several conventional protocols were evaluated, but there doesn't appear to be a direct comparison with MarShie. I feel that if a new protocol is proposed then its superiority should be quantitatively demonstrated.

2. The method proposed for removal of stripe artefact is quite simple. It is recognized that simple Fourier bandpass filtering can result in both image blurring and incomplete removal of striping artifact. Several other methods have been proposed (e.g. wavelet-FFT etc) with better performance characteristics. Why weren't one of these more sophisticated techniques used. The image after processing does appear more blurred and not all the artefact is removed.

3. What was the effect of striping on the ability for the ML model to successfully segment structures. I think it is important to demonstrate that this pre-processing step is necessary.

4. The ML algorithm is not fully described – this appears to be an important part of the pipeline, but there is very little explanation or validation data presented. This aspect requires significant expansion.

Reviewer #3 (Remarks to the Author)

Mertens et al. report a new clearing technique that is suppose to visualize subcellular details in the bone marrow. The method seems to work nicely, however this is by far not the first clearing protocol that can do it. To my knowledge Stegner et al. 2017 in Nat Commun demonstrated optical clearing of whole bones including subcellular imaging and including modeling and machine-learning based image analysis for the reconstruction of the bone and bone marrow. Furthermore Gorelashvili et al 2022 in Hematologica showed advanced analysis and modeling of cell dynamics in the bone marrow based on LSFM data and others, both not cited here. However, the fact that the here reported technique is (partly) preserving FP fluorescence is a nice feature, which unfortunately does not help much for the most prominent GFP in bone with its high autofluorescence in the green as the authors admit themselves. How well this works for a wide palette of other FPs is, unfortunately, not addressed here, however could be the main selling point.

Besides the overclaiming concerning the clearing method I appreciate the interesting results and imaging concerning the age-related changes in the bone marrow vasculature. The imaging pipeline seem to work by commercially available tools so that this could be of value for the life science community.

NCOMMS-23-09893A

Point to point response to reviewers

First of all, we would like to thank the reviewers for taking the time to read our manuscript and for their positive and constructive comments, which in our opinion helped a lot to improve the work. Below we discuss the comments point by point and explain how the reviewers' suggestions have been incorporated into the manuscript.

Reviewer #1 (Remarks to the Author)

This manuscript by Mertens et al. has reported the optimization of an optical clearing method for the 3D assessment of cellular structures in the bone marrow of mouse femurs. In the past decade, many studies have reported the development of 3D imaging strategies for intact organs of various species, including the "hard" tissues such as femurs and the skull. In light of this developing trend in the research field, the overall novelty of this study is not strong. While this work was relatively well pursued and may become publishable, several issues must be addressed.

We thank the reviewer for the assessment, we are particularly happy that he considers our work to be well pursued.

As the reviewer states, protocols for clearing hard tissues such as femurs and skull have been published previously, and we actually mentioned these protocols in the manuscript. However, our goal here was not to just achieve bone clearing, but rather to develop a protocol that that would allow quantification of single cells and even subcellular structures, such as nuclei, in the soft bone marrow of whole long bones. With already established protocols, it is possible to capture smaller areas of the bone marrow, but not the entire marrow, as it does not remain completely intact. However, imaging of the whole bone together with the bone marrow is necessary, especially for quantification and spatial analysis of rare cell types and their environment. Our requirement for image quality was that it would allow automated segmentation and quantitative analysis of small and hematopoietic cells in those bones, including the deep bone marrow regions, while preserving the tissue structure not only in homeostasis, but also in injury models. So far, no published protocol meets these criteria.

We would like to emphasize that our protocol has allowed us to gain new insights into the biology and pathophysiology of the bone marrow. As an example, we may mention the age-related changes as well as differences in the localization of myeloid cell subsets in homeostasis, or injury-related changes in myeloid cell localization. Therefore, in addition to the methodological advancement, the manuscript also contains important new biological findings. Hence, the manuscript not only proves the suitability of the method, but also provides important novel insights into bone marrow biology.

(1) The procedure of optical clearing reported in this study is a combination of several established protocols. Whether such incremental improvements would make the current method superior to others is questionable.

We thank the reviewer for pointing out that we did not explain clearly enough what led us to establish this new protocol, and what the advantages of our method are. We would like to emphasize that we did not just combine established protocols, but rather carried out our own developments, which together result in a complete analysis pipeline that meets the requirements described above.

We would also like to point out that, as previously shown, established protocols are well suited for imaging of endosteal areas or smaller parts of the bones. However, while trying to achieve clearing of complete long bones to obtain information on the bones including the marrow, we encountered several problems, which led us to develop our own strategy. In the following, we would like to explain those aspects in detail:

1. First of all, our goal was to image the whole bone and contained marrow, at cellular resolution, in order to determine spatial relationships of hematopoietic cells and quantify their location relative to each other, as well as to other structures in the bone marrow. However, established clearing methods did not preserve the complete structure of the bone marrow. They led to changes in bone marrow volume due to shrinkage, resulting in detachment of the marrow from the endost and tissue structure interruption. This was particularly evident for the FDISCO and PEGASOS clearing protocols (**Reviewer Fig. 1**). As the soft marrow shrinks in volume, the hard, mineralized cortex remains unchanged, resulting in a discrepancy in tissue size alterations evident in tissue cracks and/or detachment from the cortex at the endost. These distortions led to an incorrect mapping of the spatial relationships in the bone marrow. We are providing example images showing light sheet microscopy of the murine femur following various clearing methods for the reviewer below.

Reviewer Figure 1: Light sheet microscopy of the bone marrow in long bones after various clearing protocols. Some protocols (FDISCO, PEGASOS) distort the soft bone marrow structure. ECI clearing is not sufficient to allow light penetration into deep marrow structures.

2. Second, we aimed at developing a method, which could be applied to analyze bone injuries at cellular resolution. This is of importance because analyzing interactions between hematopoietic cells and stromal cells are crucial to understand the process of bone regeneration. In the case of a drill hole injury models, the distortion of the tissue structure by established clearing methods even led to tissue leakage out of the drill hole. This is for example evident in the macroscopic image shown after Bone CLARITY clearing (**Reviewer Fig. 2**).

Reviewer Figure 2: Example for incompatibility of tissue clearing with a bone injury model. Following the clearing procedure, bone marrow leaks out of a drill hole in the cortex. Leaked tissue is marked by the white dashed line.

As it was of importance for us to quantify the location of hematopoietic cells in relation to other cell types, those distortions were unfavorable and led to the decision to base our protocol on intramolecular epoxide linkage for tissue stabilization. This resulted in a strongly improved and reliable preservation of the tissue architecture, even after various forms of tissue injury (drill hole injury and osteotomy, shown in **Fig. 6** of the manuscript).

3. Concerning tissue clearing, we encountered the additional problem that red blood cells are present in large numbers throughout the bone marrow. This represented the main problem when we used ECI clearing, preventing imaging in deep tissue areas (**Reviewer Fig. 1**). Thus, there is a high abundance of heme, which acts a potent absorber of light, preventing deep tissue imaging. In order to remove the heme, we had to come up with a bleaching step. We succeeded using a combination of the detergent CHAPS (3-[(3-cholamidopropyl)dimethylammonio]-1-propanesulfonate) and Quadrol (N',N'-tetrakis(2-hydroxypropyl)ethylenediamine). Only this way, we could achieve a sufficient degree of clearing in order to perform deep bone marrow imaging..

Taken together, we are convinced that MarShie represents a strong improvement because it is superior to other clearing methods when analyzing the spatial context of the marrow in whole bones at a single cell level, in 3 dimensions. It fills a methodological gap in performing analyses of immune-stroma interactions in this tissue, therefore we believe it will find widespread application.

We have added text explaining the need and importance of this improvement in the introduction, and continue to revisit the topic throughout the manuscript.

(2) Related to (1), this study was primarily focused on vascular structures and their pathophysiological changes in the bone marrow. Whether those research purposes could be achieved with other established 3D imaging methods or even conventional 2D immunostaining appears unclear.

We thank the reviewer for this suggestion. Indeed, this topic is very important to us because our labs have been working on various 2D and 3D imaging methods in the past to analyze immune-stromal interactions in bone and the heterogeneity of stromal cells, including 3D (and 4D) multiphoton imaging (1-4) and 2D multiplex immunofluorescence (5).

While working on bone marrow stromal cells using histological sections, we realized that 2D methods are limited, since some stromal cells (e.g. reticular stromal cells) form long, branched extensions in all directions, which cannot be captured by those methods, preventing an accurate quantification of their sizes and cellular contacts. We emphasized this fact in the revised version of the paper by adding new data showing spatial extent of stromal cells using Prx-1 fate map reporter mice, which reveal the variety in dimensions, as well as morphological heterogeneity among those stromal cells (**new Figure 3J and K**).

Light Sheet microscopy avoids the problem of non-spherical point spread function, i.e. typically 3x larger axial than lateral resolution values, common in fluorescence microscopy (evident in both wide-field and laser scanning data, including MPM), owing to the perpendicular orientation of excitation beam path and emission detection. Additionally, the shorter excitation wavelength yields superior diffraction-limited resolution, which is a good approximation for cleared organs. The shorter excitation wavelength in LSFM leads to limited imaging depths, but excitation is performed from both sides of the sample, in contrast to 2PM. This allows better accessibility to the sample for excitation and, thus, imaging throughout the organ.

For direct comparison, inspired by the reviewer's comment, we performed two-photon imaging (2PM) of a cleared Cdh5-tdTom bone. As shown in the **new Suppl. Figure 3 and Suppl. Video 3**, we achieve a maximum depth of 400 μm . Notably, we are able to detect second harmonic generation signals in those 2PM images, demonstrating that MarShie preserves collagen structures in the bone. The possibility of label-free detection of collagen represents an interesting option in the analysis of bone biology. We have therefore decided to include these results as supplementary data in the manuscript.

Underlining our long standing interest in long bone imaging, we would like to also mention that we previously developed a lens implant compatible with 2P imaging to enable deep marrow intravital (2) imaging in the femur. While this technology enables longitudinal intravital imaging (up to months) at the same location in the deep bone marrow, it intrinsically restricts the field of view to $\sim 500 \times 500 \times 200 \mu\text{m}$, which only represents a fraction of the whole bone (2-4).

Due to the limited field of view covered by the various multiphoton methods, it has been difficult to measure the size of the central sinus in relation to the whole bone marrow. This may be the reason why profound changes in the structure of the bone marrow, such as shrinkage of the central sinus with increased age, which we demonstrate in the present manuscript, has been overlooked until now.

In summary, none of the available methods is able to capture a whole murine long bone volume in 3D at (sub)cellular resolution. This makes it hard to truly quantify the abundance of cells and their distribution over the whole marrow, which is an important question for us to address. Those methods are conceptualized for various purposes, different from highly resolved 3D imaging of whole organs. Thus, we made a conscious decision to complement our repertoire of imaging tools with light sheet microscopy. We added a paragraph to the introduction explaining our motivation in more detail.

(3) The 3D imaging results in this study were mostly achieved via the fluorescence reporter

lines, except for the whole-tissue immunostaining of anti-GFP nanobody. Such strong dependence on transgenic mouse lines limits the general utilization of the method.

We thank the reviewer for raising this important point and agree that whole-tissue immunostaining would be an advantage. We would like to point out that in vivo staining using anti-CD31 antibodies was included in the original version of the manuscript.

Following the reviewer's suggestion, we have performed additional experiments to assess the suitability of MarShie for whole-tissue immunostaining. In the revised version of the manuscript, we are now providing data in the **new Suppl. Fig 2E**, showing that whole tissue staining using a CD169-eFluor 660 antibody can be achieved in combination with MarShie. With this antibody, a clear staining is obtained throughout the bone marrow, and cellular projections of the macrophages can also be detected. This analysis reveals that the different myeloid cell types are distributed differently: while CX3CR1⁺ cells are in close contact with the vessels, CD169⁺ cells appear distributed throughout the parenchyma. These results demonstrate that antibody-based staining is compatible with the MarShie protocol and extend the applicability of the protocol beyond the use of transgenic reporter mouse strains.

(4) The manuscript boasted the implementation of machine-learning-facilitated segmentation of images. However, there needs to be solid validation of the accuracy of the algorithm.

We agree with the reviewer that the validation was missing in the previous version of the manuscript and would like to thank the reviewer for bringing up this important point. For the revised manuscript, we generated data, where we compared the performance of the algorithm to the validation of trained raters. To account for the accuracy of semantic segmentation, we used Intersection over Union (IoU, alternatively referred to as Jaccard Index or Tanimoto Coefficient) for an overlap-based metric evaluation of the algorithm (6). For that, six trained raters to generate annotations, which we and compared to the semi-automatic pixel-wise segmentation algorithm created with LABKIT. We chose two representative 3D image sections, both measuring 200 x 200 x 30 μm in the Cdh5⁺ vasculature of (i) deep marrow, located in close proximity to the central sinus and the (ii) endosteal marrow. The Intersection over union metrics between trained raters and LABKIT algorithm averaged 0,65 (0,61 – 0,68, 95% CI) and 0,68 (0,62 – 0,74, 95%, CI) for endosteal and deep marrow, respectively. In previous studies (7, 8), comparing segmentation accuracy in 3D data, IoU values between 0.6 and 0.7 have been found to be an adequate measure of successful representation of the data. In line with that, our results resemble the challenging nature of (semi-)automatic image segmentation of large, complex image volumes in addition owing to the inherent problems of interrater variances. We therefore decided to assess the performance of our approach in comparison to the variability of human annotation of the same data set, providing an intrinsically calibrated quality measure. We found that machine learning based segmentation provides a sustainable solution for interrater variance, as it allows algorithm training with data annotated by many trained raters, selecting the statistically most probable solution. The data are included in the manuscript as the **new Suppl. Fig. 5.** and we added a paragraph in the results as well as in the discussion.

Reviewer #2 (Remarks to the Author)

The authors are to be commended for their interesting study.

We thank the reviewer for his commendation.

However, I have some comments that I feel should be addressed.

1. One of the main study aims was to demonstrate that MarShie was superior to previous clearing protocols. It is stated that several conventional protocols were evaluated, but there doesn't appear to be a direct comparison with MarShie. I feel that if a new protocol is proposed then its superiority should be quantitatively demonstrated.

There were various factors that limited the use of conventional methods to clear whole bones or the marrow they contained. As outlined in the response to Reviewer 1 and exemplified by images, this did not only refer to the quality of the images generated, but also to the degree of distortion, which the bone marrow adopted during the clearing process, either by shrinking or by expansion of the tissue. This distortion is difficult to quantify, however, we think the images (**Reviewer Figure 1**) clearly demonstrate those artifacts.

In addition to tissue preservation, however, the image quality deep in the bone marrow was of particular importance to us. In order to quantify the improved optical performance of MarShie, we analyzed the SNR and SBR, profiling the signal quality in line plots, comparing PEGASOS, to our novel MarShie approach (**Reviewer Figure 3**). As the other protocols did not allow us to acquire images deep into the bone marrow, we have refrained from making a comparison here.

Both the MarShie and PEGASOS clearing methods allow imaging throughout the mouse femur, as shown in **Reviewer Figure 3**. However, next to a better contrast, MarShie leads to a superior spatial resolution as compared to PEGASOS. In endosteal areas, the spatial resolution amounts to $2.5 \pm 0.5 \mu\text{m}$ for MarShie and to $3.7 \pm 1.1 \mu\text{m}$ for PEGASOS; in the deep marrow, next to the main sinus, it amounts to $2.5 \pm 0.4 \mu\text{m}$ for MarShie and to $4.2 \pm 1.0 \mu\text{m}$ for PEGASOS.

Reviewer Figure 3: Quantitative comparison of MarShie and PEGASOS clearing methods on the example of Cdh5-tdTom femurs. **A.** Representative tdTom fluorescence images of comparable femoral areas after clearing using the MarShie or PEGASOS procedure. Scale bars are 200 μm . **B.** Representative line profiles of intensities corresponding to the yellow segments in A (left graph) and their first derivative (right graph) indicate superior contrast and spatial resolution for imaging of vascular structures when the MarShie clearing procedure is applied, as compared to the PEGASOS approach.

2. The method proposed for removal of stripe artefact is quite simple. It is recognized that simple Fourier bandpass filtering can result in both image blurring and incomplete removal of striping artifact. Several other methods have been proposed (e.g. wavelet-FFT etc) with better performance characteristics. Why weren't one of these more sophisticated techniques used. The image after processing does appear more blurred and not all the artefact is removed.

We thank the reviewer for the important question and would like to explain why we proposed a simple, robust method for stripe suppression in our LSFM images. We would like to emphasize that in our approach, we take advantage of the knowledge about the sample illumination directions in our microscope, which impact on the stripe pattern formation in the images. This aspect is emphasized in the revised manuscript.

In addition to our directional frequency suppression approach, we had tested various existing methods proposed for removing periodical patterns from images: 1. the combined wavelet-Fourier filtering proposed by (9) for unidirectional stripe pattern removal, and 2. non-

subsampling contourlet transform proposed by (10) to remove stripes in LSFM images of soft tissues, acquired with the same microscope we use.

Here, combined wavelets showed no effect on stripes at angles deviating from 0 (horizontal direction), as also mentioned in (10), due to the fact that wavelet function transformations as such are insensitive to pattern directionality. Image rotation to compensate this issue did not result in better stripe suppression, as the discrete wavelet reconstruction is based on real-valued wavelet function transformation via FFT filtering in the complex-number space. Furthermore, test images in the two publications already showed that stripes were still present. In addition, we noticed a broadening of the wavy illumination function parallel to the stripe orientation after image reconstruction, as a numerical artifact of the wavelet reconstruction.

Based on this experience, we moved to the nonsubsampling contourlet transform proposed by (10), which relies on several directional decomposition levels. Hence, variously oriented striped patterns in the image are expected to be captured and corrected by this method. Additionally, the contourlet functions used by the method are complex-valued, and thus adequate for the FFT filtering in the complex-number space. Using this method, stripe suppression succeeded in all directions, not only horizontal direction, as expected. However, as also evident in (10), additional stripes appeared after filtering by suppression of stripes at the same time. Another drawback of this method is the computation time of several minutes per image on a state-of-the-art device, as compared to 15 seconds of our method, which made the non-subsampling contourlet approach challenging for processing large image data as those acquired by LSFM.

Next, we combined the wavelet approach with self-designed FFT masks used also in our approach (along the directions of illumination in our microscope), but in this case the structural image content was not well preserved, as compared to our approach.

The results led us to design an image-processing approach to suppress the stripe signal, taking the physical imaging process into account; we thereby exploit the fact that we know the specific imaging parameters of our device. In the case of our approach, only frequencies related to the known laser beam directions are suppressed, minimizing the loss of structural information. We compared this method to the published approaches and the results are shown in the revised manuscript, in the **new Suppl. Fig. 4**. This approach allowed us to post-process the data in a time-efficient manner and improved the segmentation results as compared to raw images without de-stripping. Taken together, we present here a de-stripping solution that is tailor-made for our microscope and that saves time and resources, especially for large 3D stacks like the ones we use. We have added text better explaining our approach to the results and, also, discuss it in comparison to the other published approaches.

3. What was the effect of striping on the ability for the ML model to successfully segment structures. I think it is important to demonstrate that this pre-processing step is necessary.

We thank the reviewer for this suggestion. Indeed, the necessity for the pre-processing step was not sufficiently presented in the previous version of the manuscript. To demonstrate the importance of the de-stripping, we have added new data (**Suppl. Fig. 6**) comparing the performance of the segmentation algorithm in the same regions with and without de-stripping. From these data, it is evident that de-stripping reduces the erroneous interpretation of stripe artifacts as tissue structures. While ML-automated segmentation of larger blood vessels is readily achievable in both raw and de-striped images, stripe artifacts are misinterpreted as

small vessels in the raw images, but not in the destriped ones. This underlines the importance of de-stripping for our analyses.

4. The ML algorithm is not fully described – this appears to be an important part of the pipeline, but there is very little explanation or validation data presented. This aspect requires significant expansion.

We agree that information explaining the algorithm to the data and its validation was missing. We added text explaining the application of the algorithm for our purpose in the manuscript. The algorithm has previously been published (11), and we now added a short explanatory sentence and explicitly refer to this publication in the revised text. In order to validate the algorithm for our purpose, we compared the performance of the algorithm to the validation of trained raters and included the information in the revised manuscript (**new Suppl. Fig. 5**). Accordingly, we added text to the results section and also to the discussion of the manuscript.

Reviewer #3 (Remarks to the Author)

Mertens et al. report a new clearing technique that is suppose to visualize subcellular details in the bone marrow. The method seems to work nicely, however this is by far not the first clearing protocol that can do it.

We agree with the reviewer that protocols for bone clearing have been published previously, and we actually mentioned these protocols in the manuscript. Those methods work in regions close to the endosteum, in particular for large, compact cell types. For example, they have been successfully used for megakaryocytes. However –at least in our hands- they fail when smaller hematopoietic cells or cells with long, fine cellular extensions, such as reticular stromal cells or myeloid cells, are to be visualized deep in the marrow cavity. In addition, as laid out in the response to reviewer 1, the other protocols resulted in various degrees of tissue distortion (data provided as **Reviewer Figure 1**), which the bone marrow adopted during the clearing process. This may not be a problem if only parts of the marrow are to be analyzed and unaffected regions can be selected for analysis, but it impairs the analysis of whole bones. In addition, we aimed for a protocol which can be applied to analyze the injury site in fracture models. This requires additional care during tissue preparation. In the revised introduction, we have added text in order to emphasize the need and importance of this improvement

To my knowledge Stegner et al. 2017 in Nat Commun demonstrated optical clearing of whole bones including subcellular imaging and including modeling and machine-learning based image analysis for the reconstruction of the bone and bone marrow. Furthermore Gorelashvili et al 2022 in Hematologica showed advanced analysis and modeling of cell dynamics in the bone marrow based on LSFM data and others, both not cited here.

We are well aware of the contributions of Stegner et al., as well as Gorelashvili et al. to the field of bone marrow biology, and particular their work on megakaryocyte dynamics. Since our focus is not on the biology of megakaryocytes, we had initially, in the first version of the paper, decided to cite a different piece of work of the same group, which focused more on methodological aspects. In the revised version, we now emphasize the differences in imaging

techniques for various biological questions, thus, we refer to both Stegner et al. (Nature Communications) and Gorelashvili et al. (Hematologica) in the text.

However, the fact that the here reported technique is (partly) preserving FP fluorescence is a nice feature, which unfortunately does not help much for the most prominent GFP in bone with its high autofluorescence in the green as the authors admit themselves. How well this works for a wide palette of other FPs is, unfortunately, not addressed here, however could be the main selling point.

We would like to thank the reviewer for bringing up this important point. Following the suggestion, we have performed a set of additional experiments where we tested MarShie on reporter mouse strains for various (4 different) fluorescent proteins: 1. mtmG using Adiponectin-Cre-mtmG mice 2. Tandem red fluorescent protein (using CD19:tdRFP reporter mice, which express red fluorescence in B lymphocytes), as well as Prx1:tdRFP mice (red fluorescence in bone marrow stromal cells) 3. Monomeric Kusabira Orange (mKO1), using the Fucci mouse strains, where nuclei in the G1 phase of the cell cycle show orange fluorescence and 4. Monomeric Azami Green, where nuclei of cells in the S/G2/M phase exhibit green fluorescence. Those data are now in the **new Suppl. Figure 2A-D**.

In addition to testing the various reporter fluorophores, we have succeeded in staining whole bones, as suggested by reviewer 1. The data are also included in **Suppl. Figure 2E**.

Besides the overclaiming concerning the clearing method I appreciate the interesting results and imaging concerning the age-related changes in the bone marrow vasculature. The imaging pipeline seem to work by commercially available tools so that this could be of value for the life science community.

We thank the reviewer for this positive recognition and evaluation of our work. We toned down our statements on the clearing method in the text, and explained our motivation to establish this novel approach in more detail, so it does not represent an overstatement. We hope that we have been able to convince you of the singularity and advancement that this work represents. Indeed, we hope that MarShie will be a valuable tool for researchers working on questions related to bone and bone marrow biology.

References

1. Zehentmeier, S., K. Roth, Z. Cseresnyes, O. Sercan, K. Horn, R. A. Niesner, H. D. Chang, A. Radbruch, and A. E. Hauser. 2014. Static and dynamic components synergize to form a stable survival niche for bone marrow plasma cells. *Eur J Immunol* 44: 2306-2317.
2. Reismann, D., J. Stefanowski, R. Gunther, A. Rakhymzhan, R. Matthys, R. Nutzi, S. Zehentmeier, K. Schmidt-Bleek, G. Petkau, H. D. Chang, S. Naundorf, Y. Winter, F. Melchers, G. Duda, A. E. Hauser, and R. A. Niesner. 2017. Longitudinal intravital imaging of the femoral bone marrow reveals plasticity within marrow vasculature. *Nat Commun* 8: 2153.
3. Stefanowski, J., A. Lang, A. Rauch, L. Aulich, M. Kohler, A. F. Fiedler, F. Buttgerit, K. Schmidt-Bleek, G. N. Duda, T. Gaber, R. A. Niesner, and A. E. Hauser. 2019. Spatial

- Distribution of Macrophages During Callus Formation and Maturation Reveals Close Crosstalk Between Macrophages and Newly Forming Vessels. *Front Immunol* 10: 2588.
4. Stefanowski, J., A. F. Fiedler, M. Kohler, R. Gunther, W. Liublin, M. Tschaikner, A. Rauch, D. Reismann, R. Matthys, R. Nutzi, M. G. Bixel, R. H. Adams, R. A. Niesner, G. N. Duda, and A. E. Hauser. 2020. Limbostomy: Longitudinal Intravital Microendoscopy in Murine Osteotomies. *Cytometry A*.
 5. Holzwarth, K., R. Kohler, L. Philipsen, K. Tokoyoda, V. Ladyhina, C. Wahlby, R. A. Niesner, and A. E. Hauser. 2018. Multiplexed fluorescence microscopy reveals heterogeneity among stromal cells in mouse bone marrow sections. *Cytometry A* 93: 876-888.
 6. Reinke, A., M. D. Tizabi, M. Baumgartner, M. Eisenmann, D. Heckmann-Nötzel, A. E. Kavur, T. Rädtsch, C. H. Sudre, L. Acion, M. Antonelli, T. Arbel, S. Bakas, A. Benis, M. B. Blaschko, F. Buettner, M. J. Cardoso, V. Cheplygina, J. Chen, E. Christodoulou, B. A. Cimini, G. S. Collins, K. Farahani, L. Ferrer, A. Galdran, B. VAN Ginneken, B. Glocker, P. Godau, R. Haase, D. A. Hashimoto, M. M. Hoffman, M. Huisman, F. Isensee, P. Jannin, C. E. Kahn, D. Kainmueller, B. Kainz, A. Karargyris, A. Karthikesalingam, H. Kenngott, J. Kleesiek, F. Kofler, T. Kooi, A. Kopp-Schneider, M. Kozubek, A. Kreshuk, T. Kurc, B. A. Landman, G. Litjens, A. Madani, K. Maier-Hein, A. L. Martel, P. Mattson, E. Meijering, B. Menze, K. G. M. Moons, H. Müller, B. Nichyporuk, F. Nickel, J. Petersen, S. M. Rafelski, N. Rajpoot, M. Reyes, M. A. Riegler, N. Rieke, J. Saez-Rodriguez, C. I. Sánchez, S. Shetty, R. M. Summers, A. A. Taha, A. Tiulpin, S. A. Tsiftaris, B. VAN Calster, G. Varoquaux, Z. R. Yaniv, P. F. Jäger, and L. Maier-Hein. 2023. Understanding metric-related pitfalls in image analysis validation. *ArXiv*.
 7. Falk, T., D. Mai, R. Bensch, Ö. Çiçek, A. Abdulkadir, Y. Marrakchi, A. Böhm, J. Deubner, Z. Jäckel, K. Seiwald, A. Dovzhenko, O. Tietz, C. Dal Bosco, S. Walsh, D. Saltukoglu, T. L. Tay, M. Prinz, K. Palme, M. Simons, I. Diester, T. Brox, and O. Ronneberger. 2019. U-Net: deep learning for cell counting, detection, and morphometry. *Nat Methods* 16: 67-70.
 8. Wills, J. W., J. Robertson, P. Tournomousis, C. M. C. Gillis, C. M. Barnes, M. Minter, R. E. Hewitt, C. E. Bryant, H. D. Summers, J. J. Powell, and P. Rees. 2023. Label-free cell segmentation of diverse lymphoid tissues in 2D and 3D. *Cell Rep Methods* 3: 100398.
 9. Münch, B., P. Trtik, F. Marone, and M. Stampanoni. 2009. Stripe and ring artifact removal with combined wavelet--Fourier filtering. *Opt Express* 17: 8567-8591.
 10. Liang, X., Y. Zang, D. Dong, L. Zhang, M. Fang, X. Yang, A. Arranz, J. Ripoll, H. Hui, and J. Tian. 2016. Stripe artifact elimination based on nonsubsampling contourlet transform for light sheet fluorescence microscopy. *J Biomed Opt* 21: 106005.
 11. Arzt M, *et al.* LABKIT: Labeling and Segmentation Toolkit for Big Image Data. 2022. *Frontiers in Computer Science* 4,

REVIEWERS' COMMENTS

Reviewer #1 (Remarks to the Author):

The authors have significantly improved the quality of this work, adding a collection of new data into the original version. I think that most of my previous questions were addressed, and this revision is ready for publication.

Reviewer #2 (Remarks to the Author):

Thank you my main points have been addressed

Reviewer #3 (Remarks to the Author):

It is a pleasure to see how the manuscript has improved, both on the experimental imaging as well as the analysis side. The clearing method the authors present may become a game changer in bone LSFM if it works in other people's hands as reported.

The impact and versatility is now demonstrated by imaging various FPs so that various multi-color approaches seem to be possible.

I still think that on the citation side, the literature search could have been a bit more thorough...particularly as in the last ten years there have been multiple attempts to clear hard tissue

e.g. (Woo, J., Lee, M., Seo, J. et al. Optimization of the optical transparency of rodent tissues by modified PACT-based passive clearing. *Exp Mol Med* 48, e274 (2016)).

Also on the analysis side, many tools have been developed

e.g. (Spangenberg P, Hagemann N, Squire A, Förster N, Krauß SD, Qi Y, Mohamud Yusuf A, Wang J, Grüneboom A, Kowitz L, Korste S, Totzeck M, Cibir Z, Tuz AA, Singh V, Siemes D, Struensee L, Engel DR, Ludewig P, Martins Nascentes Melo L, Helfrich I, Chen J, Gunzer M, Hermann DM, Mosig A. Rapid and fully automated blood vasculature analysis in 3D light-sheet image volumes of different organs. *Cell Rep Methods*. 2023 Mar 17;3(3):100436. doi: 10.1016/j.crmeth.2023.100436. PMID: 37056368; PMCID: PMC10088239.)

I think this could also help the discussion, but I would leave this decision to the authors.

I have only one serious request as I am concerned about the tone of the abstract. It says: "So far, a three-dimensional analysis of the complete, intact bone marrow within the cortex of whole long bones, at subcellular resolution, has not been achieved. We established a method that stabilizes the marrow and provides subcellular resolution of fluorescent signals throughout the murine femur, enabling identification and spatial characterization of hematopoietic and stromal cell subsets." With this, the authors suggests that they are the first and only who "can" or "did" whole bone imaging. This is partly misleading as 1. There are other published bone clearing and imaging system that also can technically do it. There is nothing special here in terms of the imaging sysetm, except for the clearing that allows clonal tags. 2. To my understanding, the authors do not demonstrate quantification of the whole vascular/cellular subset in the bone, but take the easiest-to-access straight middle part of the bone. This is not to criticize biologically and does not make the study less interesting. However, I feel that the claim in the abstract is a stretch and the statement could be made more moderate.

Otherwise I recommend this work for publication.

Point by point response

NCOMMS-23-09893A

REVIEWERS' COMMENTS

Reviewer #1 (Remarks to the Author):

The authors have significantly improved the quality of this work, adding a collection of new data into the original version. I think that most of my previous questions were addressed, and this revision is ready for publication.

Response: We thank the reviewer for this positive feedback.

Reviewer #2 (Remarks to the Author):

Thank you my main points have been addressed

Response: We are delighted that we were able to address the concerns of reviewer 2 in our revision.

Reviewer #3 (Remarks to the Author):

It is a pleasure to see how the manuscript has improved, both on the experimental imaging as well as the analysis side. The clearing method the authors present may become a game changer in bone LSFM if it works in other people's hands as reported. The impact and versatility is now demonstrated by imaging various FPs so that various multi-color approaches seem to be possible.

I still think that on the citation side, the literature search could have been a bit more thorough...particularly as in the last ten years there have been multiple attempts to clear hard tissue

e.g. (Woo, J., Lee, M., Seo, J. et al. Optimization of the optical transparency of rodent tissues by modified PACT-based passive clearing. Exp Mol Med 48, e274 (2016)). Also on the analysis side, many tools have been developed e.g. (Spangenberg P, Hagemann N, Squire A, Förster N, Krauß SD, Qi Y, Mohamud Yusuf A, Wang J, Grüneboom A, Kowitz L, Korste S, Totzeck M, Cibir Z, Tuz AA, Singh V, Siemes D, Struensee L, Engel DR, Ludwig P, Martins Nascentes Melo L, Helfrich I, Chen J, Gunzer M, Hermann DM, Mosig A. Rapid and fully automated blood vasculature analysis in 3D light-sheet image volumes of different organs. Cell Rep Methods. 2023 Mar 17;3(3):100436. doi: 10.1016/j.crmeth.2023.100436. PMID: 37056368; PMCID: PMC10088239.) I think this could also help the discussion, but I would leave this decision to the authors.

I have only one serious request as I am concerned about the tone of the abstract. It says: "So far, a three-dimensional analysis of the complete, intact bone marrow within the cortex of whole long bones, at subcellular resolution, has not been achieved. We established a method that stabilizes the marrow and provides subcellular resolution of fluorescent signals throughout the murine femur, enabling identification and spatial characterization of hematopoietic and stromal cell subsets." With this, the authors suggests that they are the first and only who "can" or "did" whole bone imaging. This is partly misleading as 1. There are other published bone clearing and imaging system that also can technically do it. There is nothing special here in terms of the imaging sysetm, except for the clearing that allows clonal tags. 2. To my understanding, the authors do not demonstrate quantification of the whole vascular/cellular subset in the bone, but take the easiest-to-access straight middle part of the bone. This is not to criticize biologically and does not make the study less interesting. However, I feel that the claim in the abstract is a stretch and the statement could be made more moderate.

Otherwise I recommend this work for publication.

Response: We would like to thank reviewer 3 for the recognition of our work and for mentioning the potential that MarShie holds.

Following the suggestion of the reviewer, we have made changes to the abstract in order to avoid overstatements, the second sentence now reads: "Three-dimensional analysis of the complete, intact bone marrow within the cortex of whole long bones remains a challenge, especially at subcellular resolution."

In addition, we have mentioned PACT among the methods used for bone clearing in the introduction and cited the respective work:

"...and PACT^{15,16} was used to visualize immune cells in the BM of rodents."

We have also referenced the work on segmentation of the vasculature the reviewer suggested: "Segmentation algorithms for the quantitative analysis of vascular trees in LSFM data have been described for various organs, but not for the BM⁴⁴."